# Learning to Learn with Contrastive Meta-Objective

**Shiguang Wu[1], Yaqing Wang[2]\*, Yatao Bian[3], Quanming Yao[1,4]\***
[1]Department of Electronic Engineering, Tsinghua University
[2]Beijing Institute of Mathematical Sciences and Applications
[3] Department of Computer Science, National University of Singapore
[4]State Key laboratory of Space Network and Communications, Tsinghua University
`wsg23@mails.tsinghua.edu.cn, wangyaqing@bimsa.cn,`
`ybian@nus.edu.sg, qyaoaa@tsinghua.edu.cn`

## Abstract

Meta-learning enables learning systems to adapt quickly to new tasks, similar to humans. Different meta-learning approaches all work under/with the mini-batch episodic training framework. Such framework naturally gives the information about task identity, which can serve as additional supervision for meta-training to improve generalizability. We propose to exploit task identity as additional supervision in meta-training, inspired by the alignment and discrimination ability which is is intrinsic in human's fast learning. This is achieved by contrasting what meta-learners learn, i.e., model representations. The proposed ConML is evaluating and optimizing the contrastive meta-objective under a problem- and learner-agnostic meta-training framework. We demonstrate that ConML integrates seamlessly with existing meta-learners, as well as in-context learning models, and brings significant boost in performance with small implementation cost.

## 1 Introduction

Learning to learn, also known as meta-learning [38, 41], is a powerful paradigm designed to enable learning systems to adapt quickly to new tasks. During the meta-training phase, a meta-learner simulates adaptation (learning) across a variety of relevant tasks to accumulate knowledge on how to learn effectively. In the meta-testing phase, this learned adaptation strategy is applied to unseen tasks. The adaptation is typically accomplished by the meta-learner, which, given a set of task-specific training examples, generates a predictive model tailored to that task.

As the objective of meta-learning is to learn a meta-learner to generalize well to unseen tasks where a few labeled examples are given, the most conventional objective in meta-training follows the natural idea "train as you test" [46] to minimize the validation loss, by splitting each task into a training set (support set) to which the meta-learner would be adapted to, and a validation set (query set) to evaluate the adapted model. Beyond "train as you test", people also have introduced regularization to the meta-training objective to improve generalizability, like supervision from stronger models [52, 13, 54], or injecting global information into each task [49]. All these works are under/with the same *mini-batch episodic training* framework: sampling a batch of tasks in each episode to obtain an episodic loss to minimize.

The mini-batch episodic training framework is universal, and naturally gives the information about task identity, which can serve as additional supervision for meta-training for generalizability. Inspired by the intrinsic property of human's fast learning ability: alignment and discrimination [9, 23, 11], we hope **a meta-learner itself should be able to tell if different datasets are from the same task or different tasks** by exploiting task identity. A good learner possesses **alignment** ability

---

\*Corresponding authors.

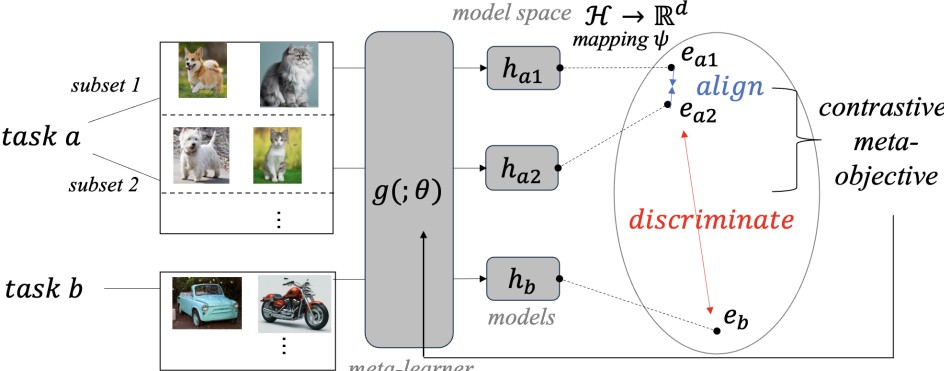

Figure 1: ConML is performing contrastive learning in model space, to make the meta-learner itself able to align information from the same task (alignment) while discriminate different tasks to improve generalizability (discrimination).

to align different partial views of a certain object, which means they can integrate various aspects or perspectives of information to form a coherent understanding [10]. This means a meta-learner should learn similar models from different datasets of the same tasks even if the data are few or noisy, benefiting the meta-testing performance through being robust to the given labeled data. **Discrimination** involves distinguishing between similar stimuli to respond appropriately only to decisive inputs. This means a meta-learner should learn different models from different tasks even if some of their inputs are similar, benefiting meta-testing performance through generalization to diverse tasks.

In this paper, we propose ConML, modifying the conventional mini-batch episodic meta-training with additional contrastive meta-objective to improve alignment and discrimination abilities of meta-learner. Similar to how contrastive learning contrasts unlabeled samples by identity, ConML contrasts the outputs of the meta-learner based on task identity. Positive pairs consist of different subsets of the same task, while negative pairs come from different tasks, with the objective of minimizing inner-task distance (alignment) and maximizing inter-task distance (discrimination). We design cheap and straightforward ways to obtain model representations for different types of meta-learners.

ConML distinguishes itself by being universal: it is **problem-agnostic**, as it is based-on mini-batch episodic training where task-identity are intrinsic information; and it is **learner-agnostic**, as we design easy-to-implement mapping functions from model to representations for different meta-learners. Additionally, it is efficient in that it requires no additional data or retraining. Existing approaches have also leveraged task-level alignment or contrastiveness as additional supervision for improved meta-learning. However, they are based on either problem-specific knowledge [52, 54, 49] or learner-specific knowledge [18, 13]. Thus, they can be improved by exploiting the problem- and learner-agnostic task identity through incorporating with ConML. Our contributions are:

- We propose to exploit task-identity as additional supervision in meta-training by emulating human cognitive alignment and discrimination abilities.

- We extend contrastive learning from the representation space in unsupervised learning to the model space in meta-learning, by designing mapping functions from models to representations for various types of meta-learners.

- We empirically show the proposed ConML universally improves the performance of various meta-learning algorithms from different categories with small implementation cost. Furthermore, we show that ConML can also improve in-context learning (ICL) as its training also follows the paradigm of learning to learn.

## 2 Preliminaries: Learning to Learn

Learning to learn, which is also known as meta-learning, focuses on improving the learning algorithm itself [38]. We focus on the most general meta-learning setting. Formally, let $g(;\theta)$ be a meta-learner

that maps a dataset $\mathcal{D}$ to a model $h$, i.e., $h = g(\mathcal{D}; \theta)$. Let $\mathcal{L}(\mathcal{D}; h)$ represent the loss when evaluating a model $h$ on a dataset $\mathcal{D}$ using a loss function $\ell(y, \hat{y})$ (e.g., cross-entropy or mean squared error). Given a distribution of tasks $p(\tau)$ for meta-training, where each task $\tau$ corresponds to a dataset $\mathcal{D}_\tau = \{(x_{\tau,i}, y_{\tau,i})\}_{i=1}^m$, the objective of meta-learning is to train $g(; \theta)$ to generalize well to unseen task $\tau'$ sampled from $p(\tau')$. During meta-testing, give an unseen task $\tau'$ with labeled dataset $\mathcal{D}_{\tau'}^{\text{tr}}$ (training set or support set) to $g(; \theta)$ to generate $h$, which is tested by another set from the same task $\mathcal{D}_{\tau'}^{\text{val}}$ (validation set or query set), i.e., evaluated by $\mathcal{L}(\mathcal{D}_{\tau'}^{\text{val}}; g(\mathcal{D}_{\tau'}^{\text{tr}}; \theta))$.

In meta-training, the meta-learner $g(; \theta)$ is optimized through a series of episodes, each consisting of a batch $\boldsymbol{b}$ of $B$ tasks, and obtains an episodic loss $\mathcal{L}_e$ to minimize. The form of $\mathcal{L}_e$ can be various, while we take the most typical validation loss as example to illustrate our method. Splitting each $\mathcal{D}_\tau$ into a training set $\mathcal{D}_\tau^{\text{tr}} = \{(x_{\tau,i}, y_{\tau,i})\}_{i=1}^n$ and a validation set $\mathcal{D}_\tau^{\text{val}} = \{(x_{\tau,i}, y_{\tau,i})\}_{i=n+1}^m$, the meta-training objective is minimizing $\mathbb{E}_{\tau \sim p(\tau)} \mathcal{L}(\mathcal{D}_\tau^{\text{val}}; g(\mathcal{D}_\tau^{\text{tr}}; \theta))$. The mini-batch episodic training with validation loss is outlined in Algorithm 1. Note that ConML relies on the mini-batch episodic framework, which is general. ConML does not rely on how the specific meta-learner measures $\mathcal{L}_e$ inside each episode. Here we take the representative validation loss as example and will discuss other forms in Section 5.

Different meta-learners implement their own specific functions within $g(; \theta)$. Popular meta-learning approaches can be broadly categorized into the following types [5]: (i) Optimization-based approaches [2, 15, 31], which focus on learning better optimization strategies for adapting to new tasks; (ii) Metric-based approaches [46, 39, 40], which leverage learned similarity metrics; and (iii) Amortization-based approaches [17, 36, 4], which aim to learn a shared representation across tasks, amortizing the adaptation process by using neural networks to directly infer task-specific parameters from the training set; (iv) Furthermore, the emerging ICL ability in

---

**Algorithm 1** Mini-Batch Episodic Training (with Validation Loss).

---
**while** Not converged **do**
    Sample a batch of tasks $\boldsymbol{b} \sim p^B(\tau)$.
    **for** All $\tau \in \boldsymbol{b}$ **do**
        Get task-specific model $h_\tau = g(\mathcal{D}_\tau^{\text{tr}}; \theta)$;
        Get validation loss $\mathcal{L}(\mathcal{D}_\tau^{\text{val}}; h_\tau)$;
    **end for**
    Get episodic loss
    $\mathcal{L}_e = \frac{1}{B} \sum_{\tau \in \boldsymbol{b}} \mathcal{L}(\mathcal{D}_\tau^{\text{val}}; g(\mathcal{D}_\tau^{\text{tr}}; \theta))$;
    Update $\theta$ by $\theta \leftarrow \theta - \nabla_\theta \mathcal{L}_e$.
**end while**

---

large language models (LLMs) can also be viewed as the consequence of meta-learning [16, 1], and ICL model is meta-learner with minimal inductive bias [53], so we will also use meta-learner $g$ to express the function of ICL. Details reformulating ICL model as meta-learner to incorporate ConML with are in Section 3.3.

# 3 Meta-Learning with ConML

Now, we introduce our ConML which equips meta-learners with the desired alignment and discrimination ability via task-level contrastive learning.

## 3.1 A General Framework

To enhance the alignment and discrimination abilities of meta-learning, we draw inspiration from Contrastive Learning (CL) [32, 8, 50]. CL focuses on learning representations that are invariant to irrelevant details while preserving essential information. This is achieved by maximizing alignment and discrimination (uniformity) in the representation space [50]. While most existing studies focus on sample-wise contrastive learning in the representation space via unsupervised learning [20, 3, 43, 8], we extend CL to the model space in meta-learning.

Specifically, we introduce contrastive meta-objective by tasks-level CL in the model space, where alignment is achieved by minimizing the inner-task distance (i.e., the distance between models trained on different subsets of the same task), and discrimination is achieved by maximizing the inter-task distance (i.e., the distance between models from different tasks). Such alignment and discrimination together form the contrastive meta-objective to optimize a meta-learner. The detailed procedures of ConML are introduced below.

*Obtaining Model Representation.* To train the meta-learner $g$, the inner-task distance $d^{\text{in}}$ and inter-task distance $d^{\text{out}}$ are measured in the output space of $g$, also referred to as the model space $\mathcal{H}$. A practical approach is to represent the model $h = g(\mathcal{D}; \theta) \in \mathcal{H}$ as a fixed-length vector $\boldsymbol{e} \in \mathbb{R}^d$, and

then compute the distances using an explicit distance function $\phi(\cdot, \cdot)$ (e.g., cosine distance). To form a learner-agnostic framework, we introduce a projection function $\psi : \mathcal{H} \to \mathbb{R}^d$ to obtain the model representations $\boldsymbol{e} = \psi(h)$. The details of $\mathcal{H}$ and $\psi$ will be specified in Section 3.3.

*Obtaining Inner-Task Distance.* Alignment is achieved by minimizing inner-task distance. During meta-training, the combined dataset $\mathcal{D}_\tau^{\text{tr}} \cup \mathcal{D}_\tau^{\text{val}}$ contains all the available information about task $\tau$. The meta-learner is expected to produce similar models when trained on any subset $\kappa$ of this dataset. Moreover, models trained on subsets should resemble the model learned from the full dataset $\mathcal{D}_\tau^{\text{tr}} \cup \mathcal{D}_\tau^{\text{val}}$. For $\forall \kappa \subseteq \mathcal{D}_\tau^{\text{tr}} \cup \mathcal{D}_\tau^{\text{val}}$, we expect $\boldsymbol{e}_\tau^\kappa = \boldsymbol{e}_\tau^*$, where $\boldsymbol{e}_\tau^\kappa = \psi(g(\kappa; \theta))$, $\boldsymbol{e}_\tau^* = \psi(g(\mathcal{D}_\tau^{\text{tr}} \cup \mathcal{D}_\tau^{\text{val}}; \theta))$. The inner-task distance $d_\tau^{\text{in}}$ for each task $\tau$ is computed as:

$$d_\tau^{\text{in}} = (1/K) \cdot \sum_{k=1}^{K} \phi(\boldsymbol{e}_\tau^{\kappa_k}, \boldsymbol{e}_\tau^*), \quad \text{s.t. } \kappa_k \sim \pi_\kappa(\mathcal{D}_\tau^{\text{tr}} \cup \mathcal{D}_\tau^{\text{val}}), \tag{1}$$

where $\{\kappa_k\}_{k=1}^K$ are $K$ subsets sampled from $\mathcal{D}_\tau^{\text{tr}} \cup \mathcal{D}_\tau^{\text{val}}$ using a specific sampling strategy $\pi_\kappa$. In each episode, given a batch $\boldsymbol{b}$ of task containing $B$ tasks, the overall inner-task distance is averaged as $d^{\text{in}} = \frac{1}{B} \sum_{\tau \in \boldsymbol{b}} d_\tau^{\text{in}}$.

*Obtaining Inter-Task Distance.* Discrimination is achieved by maximizing inter-task distance. Since the goal of meta-learning is to improve performance on unseen tasks, it is crucial for the meta-learner $g$ to generalize well across diverse tasks. Given the natural assumption that different tasks require distinct task-specific models, it is essential that $g$ can learn to differentiate between tasks—i.e., possess strong discrimination capabilities. To enhance task-level generalization, we define the inter-task distance $d^{\text{out}}$, which should be maximized to encourage $g$ to learn distinct models for different tasks. Specifically, for any two tasks $\tau \neq \tau'$ during meta-training, we maximize the distance between their respective representations, $\boldsymbol{e}_\tau^*$ and $\boldsymbol{e}_{\tau'}^*$. To make this practical within the mini-batch episodic training paradigm, we compute $d^{\text{out}}$ across a batch of tasks sampled in each episode:

$$d^{\text{out}} = (1/B(B-1)) \cdot \sum_{\tau \in \boldsymbol{b}} \sum_{\tau' \in \boldsymbol{b} \setminus \tau} \phi(\boldsymbol{e}_\tau^*, \boldsymbol{e}_{\tau'}^*). \tag{2}$$

*Training Procedure.* ConML optimizes the combination of the original episodic loss $\mathcal{L}_e$ and contrastive meta-objective $\mathcal{L}_c = d^{\text{in}} - d^{\text{out}}$:

$$\mathcal{L}_{\text{ConML}} = \mathcal{L}_e + \lambda \mathcal{L}_c \tag{3}$$

The meta-training procedure with ConML is in Algorithm 2. Note ConML is agnostic to the form of $\mathcal{L}_e$ so here we take the typical validation loss as example. Compared to Algorithm 1, ConML introduces additional computations for $\psi(g(\mathcal{D}; \theta))$ a total of $K + 1$ times per episode. However, $\psi$ is implemented as a lightweight function (e.g., extracting model weights), and $g(\mathcal{D}; \theta)$ is already part of the standard episodic training process, with multiple evaluations of $g(\mathcal{D}; \theta)$ being parallelizable. As a result, ConML incurs only a little extra cost in computation (detailed analysis is in Appendix A).

---

**Algorithm 2** Meta-Training with ConML (with Validation Loss).

---

**while** Not converged **do**
    Sample a batch of tasks $\boldsymbol{b} \sim p^B(\tau)$.
    **for** All $\tau \in \boldsymbol{b}$ **do**
        †Sample $\kappa_k$ from $\pi_\kappa(\mathcal{D}_\tau^{\text{tr}} \cup \mathcal{D}_\tau^{\text{val}})$ for $k \in \{1 \cdots K\}$;
        †Get model representation $\boldsymbol{e}_\tau^{\kappa_k} = \psi(g(\kappa_k; \theta))$;
        †Get model representation $\boldsymbol{e}_\tau^* = \psi(g(\mathcal{D}_\tau^{\text{tr}} \cup \mathcal{D}_\tau^{\text{val}}; \theta))$;
        †Get inner-task distance $d_\tau^{\text{in}}$ by (1);
        Get task-specific model $h_\tau = g(\mathcal{D}_\tau^{\text{tr}}; \theta)$;
        Get validation loss $\mathcal{L}(\mathcal{D}_\tau^{\text{val}}; h_\tau)$;
    **end for**
    †Get $d^{\text{in}} = \frac{1}{B} \sum_{\tau \in \boldsymbol{b}} d_\tau^{\text{in}}$ and $d^{\text{out}}$ by (2);
    Get loss $\mathcal{L}_{\text{ConML}}$ by (3);
    Update $\theta$ by $\theta \leftarrow \theta - \nabla_\theta \mathcal{L}$.
**end while**

---

"†" indicates additional steps introduced by ConML to Algorithm 1.

## 3.2 Provable Benefits for Generalization

Here, we provide another perspective to understand how ConML helps meta-learning. It is provable that a meta-learner which minimizes $\mathcal{L}_c$ has lower generalization error upper-bound than any other meta-learners. This means ConML serves as a 'safeguard' for the worst case of error due to finite samples in $\mathcal{D}_\tau^{\text{val}}$ in meta-testing, that can be plugged-in any meta-learners.

Following [28], the excess risk of a meta-learner $g(; \theta)$ is defined as:

$$\Delta\epsilon_{p(\tau)}(\theta) = E_{\tau \sim p(\tau)} E_{D_\tau^{tr} \sim \tau^n} E_{(x,y) \sim \tau} \ell\big(g(D_\tau^{tr}; \theta)(x), y\big) - \min_\theta E_{\tau \sim p(\tau)} \Big[ \min_{h \in H_{g(\theta)}} E_{(x,y) \sim \tau} \ell\big(h(x), y\big) \Big],$$

Table 1: Specifications of integrating ConML with different meta-learners.

| Category | Examples | Meta-learner $g(\mathcal{D};\theta)$ | Model representation $\psi(g(\mathcal{D};\theta))$ |
|---|---|---|---|
| Optimization -based | MAML, Reptile | Update model weights $\theta - \nabla_\theta \mathcal{L}(\mathcal{D}; h(;\theta))$ | Updated model weight $\theta - \nabla_\theta \mathcal{L}(\mathcal{D}; h(;\theta))$ |
| Metric -based | ProtoNet, MatchNet | Build classifier with $\{(\{f(x_i;\theta)\}_{x_i \in \mathcal{D}_j}, \text{label } j)\}_{j=1}^N$ | Concatenate $[\frac{1}{|\mathcal{D}_j|} \sum_{x_i \in \mathcal{D}_j} f(x_i;\theta)]_{j=1}^N$ |
| Amortization -based | CNPs, CNAPs | Map $\mathcal{D}$ to model weights by $H(\mathcal{D};\theta)$ | Output of hypernetwork $H(\mathcal{D};\theta)$ |
| In-context learning | In-context learning | Task-specific prediction for $x$ is given by sequential model $g([\vec{\mathcal{D}}, x];\theta)$ | $g([\vec{\mathcal{D}}, u];\theta)$, where $u$ is dummy input |

where $H_{g(\theta)}$ is the hypothesis class of $h$ given $g(;\theta)$. The value of $\Delta\epsilon_{p(\tau)}(\theta) > 0$, means the difference between expectation of validation loss between $g(;\theta)$ given finite $n$ examples per task, and the best we can find given $g$ and $p(\tau)$. First, we can find an upper bound $U_{p(\tau)}(\theta)$ for $\Delta\epsilon_{p(\tau)}(\theta)$.

**Lemma 1.** *Denote $U_{p(\tau)}(\theta) = C_1 \sqrt{\sup_{||v|| \leq 1} E_{\tau \sim p(\tau)} E_{(x,y) \sim \tau}[\langle v, g(\{(x,y)\};\theta))^2]} + C_2$. There exists positive constants $C_1, C_2$ not related with $\theta$, satisfying $\forall \theta, \Delta\epsilon_{p(\tau)}(\theta) \leq U_{p(\tau)}(\theta)$.*

Given contrastive meta-objective $\mathcal{L}_c$ as defined above, with mild assumptions and choice of $\phi$ we then have the following theorem:

**Theorem 1.** *$\forall p(\tau), U_{p(\tau)}(\theta_{\mathcal{L}_c}^*) = \min_\theta U_{p(\tau)}(\theta)$, where $\theta_{\mathcal{L}_c}^* = \arg\min_\theta \mathcal{L}_c(g(;\theta), p(\tau))$.*

This means the contrastive meta-objective can exactly serve as a surrogate objective of the worst-case meta-testing performance, as described in the above theorem. Note that this holds for any $p(\tau)$ and $g$, which indicates the problem- and learner-agnostic benefit of ConML. The proof is in Appendix B.

### 3.3 Integrating with Typical Meta-Learners

ConML is universally applicable to enhance meta-learning algorithm that follows episodic training. It does not depend on a specific form of $g$ or $\mathcal{L}_e$ and can be used alongside other forms of task-level information. Next, we provide the specifications of $\mathcal{H}$ and $\psi(g(\mathcal{D}, \theta))$ to obtain model representations for implementing ConML. We illustrate examples across different categories of meta-learning algorithms, including optimization-based, metric-based, amortization-based and ICL. They are summarized in Table 1. Appendix C provides the detailed procedures for integrating ConML with various meta-learning algorithms.

*With Optimization-Based.* The representative algorithm of optimization-based meta-learning is MAML, which meta-learns an initialization from where gradient steps are taken to learn task-specific models, i.e., $g(\mathcal{D};\theta) = h(;\theta - \nabla_\theta \mathcal{L}(\mathcal{D}; h(;\theta)))$. Since MAML directly generates the model weights, we use these weights as model representation. Specifically, the representation of the model learned by $g$ given a dataset $\mathcal{D}$ is: $\psi(g(\mathcal{D};\theta)) = \theta - \nabla_\theta \mathcal{L}(\mathcal{D}; h(;\theta))$, certain optimization-based meta-learning algorithms, such as FOMAML [15] and Reptile [31], use first-order approximations of MAML and do not strictly follow Algorithm 1 to minimize validation loss. Nonetheless, ConML can still be incorporated into these algorithms as long as they adhere to the episodic training framework.

*With Metric-Based.* Metric-based algorithms are well-suited for classification tasks. Given a dataset $\mathcal{D}$ for an $N$-way classification task, these algorithms classify based on the distances between input samples $\{\{f(x_i;\theta)\}_{x_i \in \mathcal{D}_j}\}_{j=1}^N$ and their corresponding labels, where $f(;\theta)$ is a meta-learned encoder and $\mathcal{D}_j$ represents the set of inputs for class $j$. We represent this metric-based classifier by concatenating the mean embeddings of each class in a label-aware order. For example, ProtoNet [39] computes the prototype $\boldsymbol{c}_j$, which is the mean embedding of samples in each class: $\boldsymbol{c}_j = \frac{1}{|\mathcal{D}_j|} \sum_{(x_i,y_i) \in \mathcal{D}_j} f(x_i;\theta)$. The classifier $h_\tau$ then makes predictions as $p(y = j \mid x) = \exp(-d(f(x;\theta), \boldsymbol{c}_j))/\sum_{j'} \exp(-d(f(x;\theta), \boldsymbol{c}_{j'}))$. Since the outcome model $h_\tau$ depends on $\mathcal{D}$ through $\{\boldsymbol{c}_j\}_{j=1}^N$ and their corresponding labels, the representation is specified as $\psi(g(\mathcal{D};\theta)) = [\boldsymbol{c}_1|\boldsymbol{c}_2|\cdots|\boldsymbol{c}_N]$, where $[\cdot|\cdot]$ denotes concatenation.

*With Amortization-Based.* Amortization-based approaches meta-learns a hypernetwork $H(;\theta)$ that aggregates information from $\mathcal{D}$ to task-specific parameter $\alpha$, which serves as the weights for the

main-network $h$, resulting in a task-specific model $h(;\alpha)$. For example, Simple CNAPS [4] uses a hypernetwork to generate a small set of task-specific parameters that perform feature-wise linear modulation (FiLM) on the convolution channels of the main-network. In ConML, we represent the task-specific model $h(;\alpha)$ using the task-specific parameters $\alpha$, i.e., the output of the hypernetwork $H(;\theta)$: $\psi(g(\mathcal{D};\theta)) = H(\mathcal{D};\theta)$.

*With In-Context Learning (ICL).* An ICL model makes task-specific prediction by $g([\vec{\mathcal{D}}, x]; \theta)$, where $g$ is a sequential model and $\vec{\mathcal{D}}$ is the sequentialized $\mathcal{D}$ (prompt), $[x_1, y_1, \cdots, x_m, y_m]$. The details are in Appendix D. Note that ICL does not specify an explicit output model $h(x) = g(\mathcal{D};\theta)(x)$; instead, this procedure exists only implicitly through the feeding-forward of the sequence model. Thus, obtaining the representation $\psi(g(\mathcal{D};\theta))$ by explicit model weights of $h$ is not feasible for ICL. To represent what $g$ learns from $\mathcal{D}$, we design to incorporate $\vec{\mathcal{D}}$ with a dummy input $u$, which functions as a probe and its corresponding output can be readout as representation:

$$\psi(g(\mathcal{D};\theta)) = g([\vec{\mathcal{D}}, u]; \theta), \tag{4}$$

where $u$ is constrained to be in the same shape as $x$, and has consistent value in an episode. For example, for training a ICL model on linear regression tasks we can choose $u = \mathbf{1}$, and in pretraining of LLM we can choose $u =$"*what is this task?*". The complete algorithm of ConML for training an ICL model is in Appendix C.

## 4 Empirical Studies

We provide empirical studies to understand the effect of ConML on synthetic data, which shows that learning to learn with ConML brings generalizable alignment and discrimination abilities. Code is avaliable at `https://github.com/LARS-research/ConML`.

### 4.1 Few-Shot Image Classification Performance

To show ConML brings learner-agnostic improvement, we integrate ConML into various meta-learners and evaluate the meta-learning performance on few-shot image classification problem follow existing works [46, 15, 4]. We use two few-shot image classification benchmarks: miniImageNet [46] and tieredImageNet [35], evaluating on 5-way 1-shot and 5-way 5-shot tasks.

We consider representative meta-learning algorithms from different categories, including optimization-based: MAML [15], FOMAML [15], Reptile [31]; metric-based: MatchNet [46], ProtoNet [39]; amortization-based: SCNAPs (Simple CNAPS) [4]; and the state-of-the-art ICL-based few-shot learner: CAML [14]. Note that for CAML, ConML only effect the meta-training of the ICL mode, not the pretraining of Vit feature extractor. We also incorporate ConML with meta-learners with

Table 2: Meta-testing accuracy (%) on *mini*ImageNet and *tiered*ImageNet.

| Category | Algorithm | Objective | *mini*ImageNet | | *tiered*ImageNet | |
| --- | --- | --- | --- | --- | --- | --- |
| | | | 5-way 1-shot | 5-way 5-shot | 5-way 1-shot | 5-way 5-shot |
| Optimization-Based | MAML | - | $48.75 \pm 1.25$ | $64.50 \pm 1.02$ | $51.39 \pm 1.31$ | $68.25 \pm 0.98$ |
| | | w/ ConML | $\mathbf{56.25 \pm 0.94}$ | $\mathbf{67.37 \pm 0.97}$ | $\mathbf{58.75 \pm 1.45}$ | $\mathbf{72.94 \pm 0.98}$ |
| | FOMAML | - | $48.12 \pm 1.40$ | $63.86 \pm 0.95$ | $51.44 \pm 1.51$ | $68.32 \pm 0.95$ |
| | | w/ ConML | $\mathbf{57.64 \pm 1.29}$ | $\mathbf{68.50 \pm 0.78}$ | $\mathbf{58.21 \pm 1.22}$ | $\mathbf{73.26 \pm 0.78}$ |
| | Reptile | - | $49.21 \pm 0.60$ | $64.31 \pm 0.97$ | $47.88 \pm 1.62$ | $65.10 \pm 1.13$ |
| | | w/ ConML | $\mathbf{52.82 \pm 1.06}$ | $\mathbf{67.04 \pm 0.81}$ | $\mathbf{55.01 \pm 1.28}$ | $\mathbf{70.15 \pm 1.00}$ |
| Metric-Based | MatchNet | - | $43.92 \pm 1.03$ | $56.26 \pm 0.90$ | $48.74 \pm 1.06$ | $61.30 \pm 0.94$ |
| | | w/ ConML | $\mathbf{48.75 \pm 0.88}$ | $\mathbf{62.04 \pm 0.89}$ | $\mathbf{53.29 \pm 1.05}$ | $\mathbf{67.86 \pm 0.77}$ |
| | ProtoNet | - | $48.90 \pm 0.84$ | $65.69 \pm 0.96$ | $52.50 \pm 0.96$ | $71.03 \pm 0.74$ |
| | | w/ ConML | $\mathbf{51.03 \pm 0.91}$ | $\mathbf{67.35 \pm 0.72}$ | $\mathbf{54.62 \pm 0.79}$ | $\mathbf{73.78 \pm 0.75}$ |
| Amortization-Based | SCNAPs | - | $53.14 \pm 0.88$ | $70.43 \pm 0.76$ | $62.88 \pm 1.04$ | $79.82 \pm 0.87$ |
| | | w/ ConML | $\mathbf{55.73 \pm 0.86}$ | $\mathbf{71.70 \pm 0.71}$ | $\mathbf{65.06 \pm 0.95}$ | $\mathbf{81.79 \pm 0.80}$ |
| In-Context Learning | CAML | - | $96.15 \pm 0.10$ | $98.57 \pm 0.08$ | $95.41 \pm 0.10$ | $98.06 \pm 0.10$ |
| | | w/ ConML | $\mathbf{97.03 \pm 0.10}$ | $\mathbf{98.92 \pm 0.08}$ | $\mathbf{96.56 \pm 0.09}$ | $\mathbf{98.23 \pm 0.05}$ |
| Other Objective | MELR | - | $51.33 \pm 0.73$ | $68.16 \pm 0.59$ | $54.96 \pm 0.89$ | $72.51 \pm 0.81$ |
| | | w/ ConML | $\mathbf{53.56 \pm 1.02}$ | $\mathbf{70.04 \pm 0.95}$ | $\mathbf{57.06 \pm 0.90}$ | $\mathbf{74.21 \pm 0.78}$ |
| | LastShot | - | $64.80 \pm 0.20$ | $81.65 \pm 0.14$ | $69.37 \pm 0.23$ | $85.36 \pm 0.16$ |
| | | w/ ConML | $\mathbf{66.24 \pm 0.72}$ | $\mathbf{83.29 \pm 0.45}$ | $\mathbf{71.82 \pm 0.70}$ | $\mathbf{87.05 \pm 0.49}$ |

improved meta-training objective as discussed in Section 5, including: MELR [13] and LastShot [54]. We evaluate the meta-learning performance of each algorithm in its original form (w/o ConML) and after incorporating ConML into the training process (w/ ConML). The implementation of ConML follows the general procedure described in Algorithm 2 and the specification for corresponding category in Section 3.3.

Table 2 shows the results on *mini*ImageNet and *tiered*ImageNet respectively. We uses a common configuration for ConML's hyperparameter for all meta-learners: task batch size $B = 32$, inner-task sampling $K = 1$, and $\pi_\kappa(\mathcal{D}_\tau^{\text{tr}} \cup \mathcal{D}_\tau^{\text{val}}) = \mathcal{D}_\tau^{\text{tr}}$, $\phi(a, b) = 1 - a \cdot b / \|a\| \|b\|$ and $\lambda = 0.1$. Other hyperparameters related to model architecture and training procedure remain consistent with the original meta-learners'. This demonstrates boosted performance can be brought even without specific hyperparameter tuning for different meta-learners. The performance improvement demonstrates that ConML offers universal improvements across different meta-learning algorithms. Note that performance between different algorithms are not comparable. We also show ConML's consistent benefit on different sizes of backbones in Appendix F.

## 4.2 Cross-Domain Few-Shot Image Classification Performance

To show that ConML is problem-agnostic, we provide learner-agnostic improvement on large-scale cross-domain few-shot image classification problem, obtained on META-DATASET [44]. Table 3 shows the results. The backbone and setting of P>M>F [22] is different with the other baselines [44], so they are not comparable across baselines. ConML is introduced with the same setting as Section 4.1 (inner-task sampling $K = 1$ and $\pi_\kappa(\mathcal{D}_\tau^{\text{tr}} \cup \mathcal{D}_\tau^{\text{val}}) = \mathcal{D}_\tau^{\text{tr}}$, $\phi(a, b) = 1 - a \cdot b / \|a\| \|b\|$ (cosine distance) and $\lambda = 0.1$.). Note that for P>M>F, ConML is integrated into the meta-training phase, and all other phases remain unchanged. As shown, ConML brings consistent improvement.

Table 3: Cross-domain results on META-DATASET (accuracy (%)).

| Baseline | MatchNet | | ProtoNet | | fo-MAML | | fo-Proto-MAML | | P>M>F | |
|---|---|---|---|---|---|---|---|---|---|---|
| ConML | w/o | w/ | w/o | w/ | w/o | w/ | w/o | w/ | w/o | w/ |
| ILSVRC | 45.0 | **51.1** | 50.5 | **52.3** | 45.5 | **54.1** | 49.5 | **54.3** | 77.0 | **78.6** |
| Omniglot | 52.2 | **54.6** | 59.9 | **61.2** | 55.5 | **63.7** | 63.3 | **69.8** | 91.7 | **93.3** |
| Aircraft | 48.9 | **51.5** | 53.1 | **54.9** | 56.2 | **64.9** | 55.9 | **61.5** | 89.7 | **91.1** |
| Birds | 62.2 | **66.8** | 68.7 | **68.9** | 63.6 | **69.9** | 68.6 | 68.6 | 92.9 | **94.0** |
| Textures | 64.1 | **67.6** | 66.5 | **68.4** | 68.0 | **72.3** | 66.4 | **69.4** | 86.9 | **87.5** |
| Quick Draw | 42.8 | **46.7** | 48.9 | **50.0** | 43.9 | **48.5** | 51.5 | **53.1** | 80.2 | **83.3** |
| Fungi | 33.9 | **36.4** | 39.7 | **40.9** | 32.1 | **40.6** | 39.9 | **43.7** | 78.2 | **80.1** |
| VGG Flower | 80.1 | **84.9** | 85.2 | **88.0** | 81.7 | **90.4** | 87.1 | **91.0** | 95.7 | **96.8** |
| Traffic Signs | 47.8 | **49.5** | 47.1 | **48.6** | 50.9 | **52.2** | 48.8 | **51.5** | 89.8 | **94.0** |
| MS COCO | 34.9 | **40.1** | 41.0 | **42.4** | 35.3 | **43.5** | 43.7 | **48.9** | 64.9 | **68.4** |

## 4.3 Model Analysis

We show ConML does not require much efforts on tuning hyperparameters. Furthermore, better performance can be obtained through hyperparameter optimization for specific meta-learners. In this Section we show the impact of key ConML settings: (1) the number of subset samples $K$, which influences the model's complexity, and (2) the contrastive loss, including the distance function $\phi$, the weighting factor $\lambda$, and the use of InfoNCE as a replacement for $(d^{\text{in}} - d^{\text{out}})$.

### 4.3.1 Effect of the Number of Subset Samples $K$

Table 4 presents the results of varying the number of subset samples $K$. Starting from $K = 1$, we observe moderate performance growth as $K$ increases, while memory usage grows linearly with $K$. Notably, there is a significant discrepancy in both performance and memory (approximately $\sim 2\times$) between the configurations without ConML and with $K = 1$. However, $K$ has a negligible impact on time efficiency, assuming sufficient memory, as the processes are independent and can be executed in parallel.

### 4.3.2 The Design of Contrastive Loss

Here, we explore various design factors of the contrastive loss. ConML optimizes the following objective: $\mathcal{L}_{\text{ConML}} = \mathcal{L}_e + \lambda \mathcal{L}_c$. In the previous sections, to highlight our motivation and perform a decoupled analysis, we used the naive contrastive loss $\mathcal{L}_c = d^{\text{in}} - d^{\text{out}}$, with the natural cosine distance $\phi(x, y)$. Here, we consider distance function $\phi$ as Euclidean distance, contrastive loss $\mathcal{L}_c$

in the form of InfoNCE [32], varying contrastive weight $\lambda$ in a wide range. More details are in Appendix G.1.

Table 4: The effect of subset sampling number $K$.

| | | w/o | K=1 | 4 | 16 | 32 |
|---|---|---|---|---|---|---|
| MAML w/ ConML | Acc.(%) | 48.75 | 56.25 | 56.08 | **57.59** | 57.33 |
| | Mem.(MB) | 1331 | 2801 | 3011 | 4103 | 5531 |
| | Time (relative) | 1× | 1.1× | 1.1× | 1.1× | 1.1× |
| ProtoNet w/ ConML | Acc.(%) | 48.90 | 51.03 | 52.04 | 52.34 | **52.48** |
| | Mem.(MB) | 7955 | 14167 | 15175 | 19943 | 26449 |
| | Time (relative) | 1× | 1.2× | 1.2× | 1.2× | 1.2× |

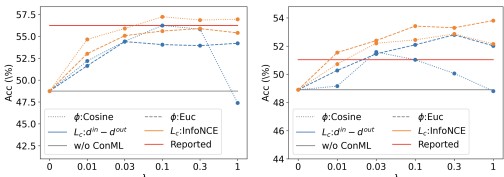

(a) MAML w/ ConML. (b) ProtoNet w/ ConML.

Figure 2: The effect of distance function $\phi$, contrastive loss form $\mathcal{L}_c$, contrastive weight $\lambda$.

Figure 2 presents the results. We observe that ConML can significantly improve the performance of meta-learners across a considerable range of $\lambda$, though setting $\lambda$ too high can lead to model collapse by overshadowing the original meta-learning objective. The choice of distance function varies between algorithms, with some performing better with specific functions. Additionally, InfoNCE outperforms the naive contrastive strategy, offering greater potential and reduced sensitivity to hyperparameters.

These findings suggest that we may not have yet reached the full potential of ConML, and there are several promising directions for further improvement. For instance, refining batch sampling strategies to account for task-level similarities or developing more advanced subset-sampling methods could enhance performance further [26, 47, 48]. We also notice that the matching between the chosen distance metric and model representation is the key to success. We can find that Euclidean distance performs much better than cosine in ProtoNet, since ProtoNet makes classification with Euclidean distance, and ConML contrasts the classifier's weights describing the model's behavior more precisely. Although cosine works generally, it would be interesting to tailor distance metrics for various parameter types (e.g., classifiers, MLPs, CNNs, GNNs, Transformers). It is also worth noting that a more delicate choice of distance metric is implied by Theorem 1. See details in Appendix B.

In Appendix E, we provide empirical results under synthetic dataset to understand (i) learning to learn with ConML brings generalizable alignment and discrimination abilities; and (ii) alignment enhances fast-adaptation and discrimination enhances task-level generalizability.

## 4.4 ICL Performance

Following [16], we investigate ConML on ICL by learning to learn synthetic functions including linear regression (LR), sparse linear regression (SLR), decision tree (DT) and 2-layer neural network with ReLU activation (NN). We train the GPT-2 [34]-like transformer for each function with ICL and ICL w/ ConML respectively and compare the inference (meta-testing) performance. We follow the same model structure, data generation and training settings [16]. More implementation details are provided in Appendix G.2.

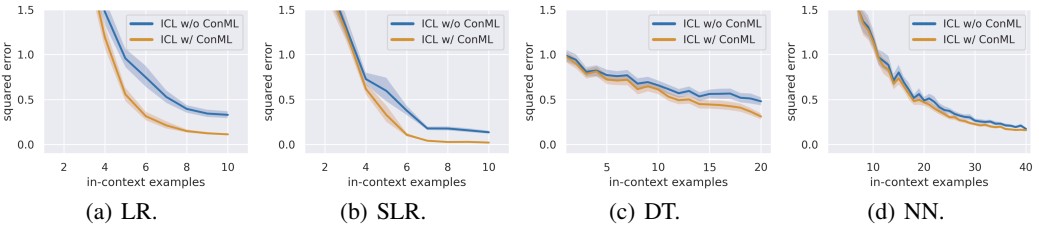

(a) LR. (b) SLR. (c) DT. (d) NN.

Figure 3: Varying the number of in-context examples during inference of ICL.

Figure 3 shows the performance, where ICL w/ ConML always makes more accurate predictions than ICL w/o ConML. Table 5 shows the two values to show the effect ConML brings to ICL: *Rel. Min. Error* is ICL w/ ConML's minimal inference error given different number of examples, divided by ICL's; and *Shot Spare* is when ICL w/ ConML obtain an error no larger than ICL's minimal error, the difference between the corresponding example numbers. One can observe significant improvement.

Table 5: Relative minimal error (Rel. Min. Error) and spared example number to reach the same error (Shot Spare) comparing ICL w/ and w/o ConML.

| Function (max prompt len.) | LR (10 shot) | SLR (10 shot) | DT (20 shot) | NN (40 shot) |
|---|---|---|---|---|
| Rel. Min. Error | $0.42 \pm 0.09$ | $0.49 \pm .06$ | $0.81 \pm 0.12$ | $0.74 \pm 0.19$ |
| Shot Spare | $-4.68 \pm 0.45$ | $-3.94 \pm 0.62$ | $-4.22 \pm 1.29$ | $-11.25 \pm 2.07$ |

The effect of ConML on ICL is without loss of generalizability to real-world applications like pretrained LLMs.

# 5   Relation with Existing Works

**Task-Identity in Meta-Learning**   There are existing works attempt to leverage task-identity information into meta-learning, but no "counterpart" for ConML. We discuss them in two categories: (i) The first category is using problem-specific information, while ConML uses problem-agnostic information thus can be plugged-in these methods and brings improvement. They primarily focusing on few-shot image classification problem [12, 19, 33], and require a static pool of base classes for meta-training and class-level alignment [51, 13, 54, 42, 49]. These problem-specific approaches are limited by their focus on few-shot classification and cannot effectively handle dynamic or diverse tasks, nor can they generalize to other meta-learning problems beyond classification. As such, they are not directly comparable with ConML. However, ConML can be integrated into these methods. Though they introduce new objectives other than validation loss by additional modules or steps, but they all work under/with the general mini-batch episodic training, either by replacing the steps to obtain $\mathcal{L}_e$ in Algorithm 1 with their steps to obtain episodic loss, or introduce additional steps outside Algorithm 1. We demonstrate in Section 4.1 that incorporating ConML leads to performance gains. (ii) The second category includes works that are learner-specific but not problem-specific. For example, [18] and [27] explore contrastive representations for neural processes. However, their methods are tightly coupled with specific meta-learners that involve explicit model representation vectors, which can be seen as special cases of ConML within amortization-based meta-learners.

**Contrastive Learning with Meta-Learning**   Some studies involve both meta-learning and contrastive learning as key components, but they are not directly related to ConML. [30] reformulates contrastive learning through meta-learning for better unsupervised learning, while [55] proposes an optimization-based meta-learner inspired by contrastive Hebbian learning in biology, which is not related to the contrastive learning used in unsupervised learning. [25] introduces contrastive set representations for unsupervised meta-learning but does not integrate them with the general meta-learning framework or model.

# 6   Conclusion, Limitations and Discussion

In this work, we propose ConML, a universal, learner-agnostic contrastive meta-learning framework that emulates the alignment and discrimination capabilities integral to human fast learning, achieved through task-level contrastive learning in the model space. ConML can be seamlessly integrated with meta-training procedure of existing meta-learners, by modifying the conventional mini-batch episodic training, and we provide specific implementations across a wide range of meta-learning algorithms. Empirical results show that ConML consistently and significantly enhances meta-learning performance by improving the meta-learner's fast-adaptation and task-level generalization abilities. Additionally, we explore in-context learning by reformulating it within the meta-learning paradigm, demonstrating how ConML can be effectively integrated to boost performance.

The primary contribution of ConML is offering a universal framework built and on the general meta-learning setting and training procedure, to reflects the inherency of alignment and discrimination as meta-training objective and the efficacy of learning to learn with contrasting model representation. The cost of ConML is additional training cost, as dicussed in Section 4.3.1, which is moderate but indelible under such framework. The current implementation of ConML is relatively primitive, as discussed in Section 4.3.2, there are many directions for further improvement, such as optimizing sampling strategies, task-scheduling, refining the contrastive strategy and tailoring model representations and distance metrics.

## Acknowledgment

Y. Wang is sponsored by Beijing Nova Program. Q. Yao is supported by National Natural Science Foundation of China (under Grant No. 92270106) and Beijing Natural Science Foundation (under Grant No. 4242039). Y. Bian is supported by the National University of Singapore SoC (grant no: A-0010308-00-00).

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

# A  Complexity Analysis

We compare the relative complexity of computing the original meta-objective and the additional contrastive objective introduced by ConML.

## A.1  ICL

For ICL model like LLM, ConML does not obtain model representation by explicit model parameters, but by simply adding an additional token to the forward-pass ($u$ in (4)). Which means pretraining a LLM with ConML only requires $K/n$ ($K$: subset sampling number, $n$: average sentence length, typically $K/n << 1$) times computation comparing with pretraining a LLM without ConML, regardless of the model size.

## A.2  Typical Meta-Learners

For typical meta-learners, denote the model representation has $d$ parameters, i.e., $e \in R^d$. We discuss about the complexity of the original computation path $h = g(D; \theta) \to \mathcal{L}_e$ and the additional computation path $h = g(D; \theta) \to \psi(h) \to \mathcal{L}_c$ introduced by ConML. We consider giving a single input sample in 1-d vector, the complexity $O_{h \to \mathcal{L}_e}$ to calculate the loss $\mathcal{L}_e = \ell(h(x), y)$, and the complexity $O_{h \to \psi(h) \to \mathcal{L}_c}$ to calculate $\mathcal{L}_c = d(\psi(h), \psi(h))$.

- For optimization-based, e.g., MAML, we have $d = |\theta| = |h|$. We consider $h$ as a $l$-layer MLP, with each average layer size $(|h|/l)^{1/2} * (|h|/l)^{1/2}$. With a single input sample, $O_{h \to \mathcal{L}_e} = O(l * (|h|/l)^{3/2})$, $O_{h \to \psi(h) \to \mathcal{L}_c} = O(|d|) + O(|d|) = O(|h|)$. While $l << |\theta|$, we have $O_{h \to \mathcal{L}_e} > O_{h \to \psi(h) \to \mathcal{L}_c}$.

- For metric-based, e.g., ProtoNet, $\theta$ corresponds to the parameter in feature extractor like CNN. $h$ is the final classifier which makes prediction by Euclidean distance, which can be viewed as a linear classifier with parameter size in $N * |h|/N$. $d$ equals to the N (ways per task) times the dimension of the embedding of a each sample $|h|/N$, $d = |h|$. A sample $x \in R^{|h|/N}$. We have $O_{h \to \mathcal{L}_e} = O(|h|^2/N)$, $O_{h \to \psi(h) \to \mathcal{L}_c} = O(d) + O(d) = O(|h|)$. With $|h| >> N$, $O_{h \to \mathcal{L}_e} > O_{h \to \psi(h) \to \mathcal{L}_c}$.

- For amortization-based, e.g., Simple CNAPs. Denote $q$ as the dimension of task-adaptive parameters generated by hypernetwork $H_\theta(D)$. $|d| = q$, $O_{h \to \psi(h) \to \mathcal{L}_c} = O(d) + O(d) = O(q)$. Consider l layers in main-network modulated by $H_\theta(D)$ feature-wisely, the $q = \sqrt{|h|/l}$, $O_{h \to \mathcal{L}_e} = O(l * (|h|/l)^{3/2}) + O(q/l)$. With $l << |h|$, we have $O_{h \to \mathcal{L}_e} > O_{h \to \psi(h) \to \mathcal{L}_c}$.

To summarize, for ICL model like LLMs, the complexity to pretrain with ConML is $\frac{n+K}{n} \approx 1$ times the complexity to pretrain without ConML. For typical meta-learners, the additional introduced objective in ConML is comparably less than the complexity of the original meta-training objective, which verifies the empirical computation cost presented in Table 4 in main text.

# B Provable Benefits for Generalization

Here we first provide the proof of Lemma 1 which shows $U_{p(\tau)}(\theta)$ is an upper bound of the excess risk of meta-learning $\Delta\epsilon_{p(\tau)}(\theta)$, and then the proof of Theorem 1 which shows minimizing contrastive meta-objective is minimizing $U_{p(\tau)}(\theta)$.

We need two preliminary results:

**Lemma 2** (Upper Bound from [28]). $\forall\theta$, $\Delta\epsilon_{p(\tau)}(\theta) \leq U_{p(\tau)}(\theta)$, where

$$\Phi_{p(\tau)}(\theta) = C_1\sqrt{E_{\tau\sim p(\tau)}E_{(x,y)\sim\tau}[||\psi(g((x,y);\theta))||^2]} + C_2,$$

with $C_1, C_2 > 0$, $\frac{dC_1}{dg} = \frac{dC_2}{dg} = 0$.

**Lemma 3** (Universal Approximation of MLP from [21]). *Let $\sigma : \mathbb{R} \to \mathbb{R}$ be a non-constant, bounded, and continuous function. Let $K$ be a compact subset of $\mathbb{R}^n$. The set of real-valued continuous functions on $K$ is denoted by $C(K)$. For any function $f \in C(K)$ and for any error tolerance $\delta > 0$, there exists an integer $N$ (the number of neurons in the hidden layer), and real constants $v_i, b_i \in \mathbb{R}$ and vectors $w_i \in \mathbb{R}^n$ for $i = 1, \ldots, N$, such that we can define the MLP output function $F : K \to \mathbb{R}$ as:*

$$F(x) = \sum_{i=1}^{N} v_i\sigma(w_i^T x + b_i)$$

*which satisfies $\forall x \in K$ and $|f(x) - F(x)| < \delta$.*

## B.1 Proof of Lemma 1

*Proof.* We use the Universal Approximation of MLP to bound the $\psi$ in $\Phi_{p(\tau)}(\theta)$.

For the model space $\mathcal{H}$ (recall $h = g(\mathcal{D};\theta) \in \mathcal{H}$) is closed and bounded by the meta-learner's hypothesis, which is a compact set on $\mathbb{R}^{|\mathcal{H}|}$, we can apply Theorem 3 on $\psi$ in $\Phi_{p(\tau)}(\theta)$. There exists two-layer MLP $F$ with a nonlinear activation function $\sigma$. Then

$$\forall X_g \in \mathcal{H}, \quad |\psi(X_g) - F(X_g)| < \delta, \tag{5}$$

where $X_g$ represents the parameter vector learned by $g(;\theta)$ from sample $(x, y)$.

As $X_g \in \mathcal{H}$, $F(X_g) \in \mathbb{R}$ and $||F(X_g)|| = ||\psi(X_g)|| \leq ||X_g||$ by definition, we have

$$\exists w_1, \cdots, w_N \in \mathbb{R}^{|\mathcal{H}|}, \sum_{k=1}^{N} ||w_k||^2 \leq N, F(X_g) = \sum_{k=1}^{N} \sigma(\langle w_k, X_g\rangle)$$

As $\delta$ can be arbitrarily small by selecting large enough $N$, we approximately rewrite (5) as $\psi(X_g) = F(X_g)$ which is not completely rigorous but no harm to our proof. We have

$$E_{\tau\sim p(\tau)}E_{(x,y)\sim\tau}[||\psi(g((x,y);\theta))||^2] = E_{X\sim P_{g(\theta)(\tau,x,y)}}[\sum_k \alpha(\langle w_k, X_g\rangle)^2],$$

where $P_{g(\theta)(\tau,x,y)}$ is the distribution of $X_g$ output by meta-learner $g(;\theta)$ on defined $p(\tau), x, y$.

Let the activation function $\sigma$ has Lipschitz constant $L_\sigma$ and $\sigma(0) = 0$. We have

$$E_{X\sim P_{g(\theta)(\tau,x,y)}}[\sum_k \alpha(\langle w_k, X_g\rangle)^2] \leq L_\sigma^2 \sum_{k=1}^{N} ||w_k||^2 E_{X\sim P_{g(\theta)(\tau,x,y)}}[\langle \frac{w_k}{||w_k||}, X_g\rangle^2]$$

$$\leq L_\sigma^2 N \sup_{||v||\leq 1} E_{X\sim P_{g(\theta)(\tau,x,y)}}[\langle v, X\rangle^2].$$

So we have

$$\Phi_{p(\tau)}(\theta) = C_1\sqrt{E_{\tau\sim p(\tau)}E_{(x,y)\sim\tau}[||\psi(g((x,y);\theta))||^2]} + C_2$$

$$\leq C_1\sqrt{L_\sigma^2 N \sup_{||v||\leq 1} E_{\tau\sim p(\tau)}E_{(x,y)\sim\tau}[\langle v, g(\{(x,y)\};\theta)\rangle^2]} + C_2 = U_{p(\tau)}(\theta).$$

So $U_{p(\tau)}(\theta)$ is a upper bound of he the excess risk of a meta-learner $\Delta\epsilon_{p(\tau)}(\theta)$. $\square$

## B.2 Proof of Theorem 1

We need assuming $\forall \mathcal{D}, ||g(\mathcal{D}; \theta)|| = 1$, and choosing the distance function $\phi$ in $\mathcal{L}_c$ to be $\phi(a, b) = -\frac{a \cdot b}{||a|| ||b||}$ for $d^{in}$ and $\phi(a, b) = -(\frac{a \cdot b}{||a|| ||b||})^2$ for $d^{out}$. We will discuss about the rationality of such assumption and choice after proof.

*Proof.* On the one hand, we have

$$U_{p(\tau)}(\theta) = C_1 \sqrt{L_\sigma^2 N \sup_{||v|| \leq 1} E_{\tau \sim p(\tau)} E_{(x,y) \sim \tau}[\langle v, g(\{(x,y)\}; \theta) \rangle^2]} + C_2$$

$$= C_1 \sqrt{L_\sigma^2 N \sup_{||v|| \leq 1} E_{X \sim P_{g(\theta)(\tau,x,y)}}[\langle v, X \rangle^2]} + C_2, \ s.t. ||X|| = 1$$

$$\geq C_1 \sqrt{\frac{L_\sigma^2 N}{|\mathcal{H}|}} + C_2,$$

where the minimum of $U_{p(\tau)}(\theta)$

$$U_{p(\tau)}^*(\theta) = C_1 \sqrt{\frac{L_\sigma^2 N}{|\mathcal{H}|}} + C_2$$

is achieved if and only if

$$\sup_{||v|| \leq 1, s.t. ||X|| = 1} E_{X \sim P_{g(\theta)(\tau,x,y)}}[\langle v, X \rangle^2] = \frac{1}{|\mathcal{H}|},$$

which is achieved if and only if

$$\forall ||X|| = 1, P_{g(\theta)(\tau,x,y)}(X) = \frac{\Gamma(\frac{|\mathcal{H}|}{2})}{2\pi^{\frac{|\mathcal{H}|}{2}}},$$

i.e., $X$ uniformly distribute on the unit sphere in $\mathbb{R}^{|\mathcal{H}|}$.

On the other hand, by definition we have

$$\mathcal{L}_c = d^{in} - d^{out}$$

$$= E_{X_{\tau,\kappa} \sim P_{g(\theta,\pi_\kappa)(\tau,x,y,\kappa)}} \left[ -\frac{\langle X_{\tau,\kappa}, X_{\tau,\kappa'} \rangle}{||X_{\tau,\kappa}|| ||X_{\tau,\kappa'}||} + \left( \frac{\langle X_\tau, X_{\tau'} \rangle}{||X_\tau|| ||X_{\tau'}||} \right)^2 \right]$$

$$= E_{\tau \sim P_{g(\theta)(\tau)}} \left[ E_{X_{\tau,\kappa} \sim P_{g(\theta,\pi_\kappa)(x,y,\kappa|\tau)}} \left[ -\frac{\langle X_{\tau,\kappa}, X_{\tau,\kappa'} \rangle}{||X_{\tau,\kappa}|| ||X_{\tau,\kappa'}||} \right] + \left( \frac{\langle X_\tau, X_{\tau'} \rangle}{||X_\tau|| ||X_{\tau'}||} \right)^2 \right]$$

$$= - E_{\tau \sim P_{g(\theta)(\tau)}, X_{\tau,\kappa} \sim P_{g(\theta,\pi_\kappa)(x,y,\kappa|\tau)}} \left[ \frac{\langle X_{\tau,\kappa}, X_{\tau,\kappa'} \rangle}{||X_{\tau,\kappa}|| ||X_{\tau,\kappa'}||} \right] + E_{\tau \sim P_{g(\theta)(\tau)}} \left[ \left( \frac{\langle X_\tau, X_{\tau'} \rangle}{||X_\tau|| ||X_{\tau'}||} \right)^2 \right].$$

For arbitrary subset sampling strategy $\pi_\kappa$, we have

$$\min_{\theta, s.t. ||X|| = 1x} \mathcal{L}_c \geq -1 + \frac{1}{|\mathcal{H}|},$$

where the minimum of $\mathcal{L}_c$

$$\mathcal{L}_c^* = -1 + \frac{1}{|\mathcal{H}|},$$

is achieved if and only if

$$E_{\tau \sim P_{g(\theta)(\tau)}, X_{\tau,\kappa} \sim P_{g(\theta,\pi_\kappa)(x,y,\kappa|\tau)}} \left[ \frac{\langle X_{\tau,\kappa}, X_{\tau,\kappa'} \rangle}{||X_{\tau,\kappa}|| ||X_{\tau,\kappa'}||} \right] = 1,$$

$$E_{\tau \sim P_{g(\theta)(\tau)}} \left[ \left( \frac{\langle X_\tau, X_{\tau'} \rangle}{||X_\tau|| ||X_{\tau'}||} \right)^2 \right] = \frac{1}{|\mathcal{H}|},$$

which is achieved if and only if

$$\forall ||\tau|| = 1, P_{g(\theta)}(\tau) = \frac{\Gamma(\frac{|\mathcal{H}|}{2})}{2\pi^{\frac{|\mathcal{H}|}{2}}}, P_{g(\theta,\pi_\kappa)(x,y,\kappa|\tau)}(X \mid \tau) = \delta_\tau(X),$$

where $\delta_\tau$ is the Dirac-delta function centered on $\tau$. Then $\forall ||X|| = 1$,

$$P_{g(\theta,\pi_\kappa)(\tau,x,y,\kappa)}(X) = \int_{||\tau||=1} P_{g(\theta)}(\tau) P_{g(\theta,\pi_\kappa)(x,y,\kappa|\tau)}(X \mid \tau) \, d\tau$$

$$= \int_{||\tau||=1} \frac{\Gamma(\frac{|\mathcal{H}|}{2})}{2\pi^{\frac{|\mathcal{H}|}{2}}} \delta_\tau(X) \, d\tau$$

$$= \frac{\Gamma(\frac{|\mathcal{H}|}{2})}{2\pi^{\frac{|\mathcal{H}|}{2}}}$$

Combining both hands, we have

$$\theta^*_{\mathcal{L}_c} = \arg\min_\theta \mathcal{L}_c(g(;\theta), p(\tau))$$

$$\Rightarrow \forall ||X|| = 1, P_{g(\theta,\pi_\kappa)(\tau,x,y,\kappa)}(X) = \frac{\Gamma(\frac{|\mathcal{H}|}{2})}{2\pi^{\frac{|\mathcal{H}|}{2}}}$$

$$\Leftrightarrow U_{p(\tau)}(\theta^*_{\mathcal{L}_c}) = C_1 \sqrt{\frac{L_\sigma^2 N}{|\mathcal{H}|}} + C_2 = \min_\theta U_{p(\tau)}(\theta)$$

$\square$

Now we discuss the implication of assuming $\forall \mathcal{D}, ||g(\mathcal{D}; \theta)|| = 1$, and choosing the distance function $\phi$ in $\mathcal{L}_c$ to be $\phi(a, b) = -\frac{a \cdot b}{||a|| ||b||}$ for $d^{in}$ and $\phi(a, b) = -(\frac{a \cdot b}{||a|| ||b||})^2$ for $d^{out}$. $||g(\mathcal{D}; \theta)|| = 1$ can be viewed as a regularization on model weights of $h$ that prevents trivial solution and enhances generalization.

The choice of $\phi$ is more inspiring: comparing with ordinary cosine distance for both $d^{in}$ and $d^{out}$, above form modifies $d^{out}$. For maximizing $d^{out}$, if we also choose ordinary cosine distance $\phi(a, b) = -\frac{a \cdot b}{||a|| ||b||}$, given a pair of tasks, it is optimized if and only if the two tasks are "opposite", i.e., still coupled, while the modified $-(\frac{a \cdot b}{||a|| ||b||})^2$ is maximized if and only if the two tasks are "orthogonal", i.e., decoupled. The latter form is preferred to be more reasonable. However, this understanding has only come to our mind after our work, so we have not implemented in our experiments. This could be studied as a future direction.

# C   Specifications of Meta-Learning with ConML

Here, we provide the specific algorithm process of representative implementation ConML, including the universal framework of ConML (Algorithm 3), the most efficient implementation of ConMLwith $K = 1$ and $\pi_\kappa(\mathcal{D}^{\mathrm{tr}}_\tau \cup \mathcal{D}^{\mathrm{val}}_\tau) = \mathcal{D}^{\mathrm{tr}}_\tau$ (Algorithm 4), training ICL model with ConML (Algorithm 5), MAML w/ ConML (Algorithm 6), Reptile w/ ConML (Algorithm 7), SCNAPs w/ ConML (Algorithm 8), ProtoNet w/ ConML (Algorithm 9).

---

**Algorithm 3** ConML.

---

**Input:** Task distribution $p(\tau)$, batch size $B$, inner-task sample times $K$ and sampling strategy $\pi_\kappa$.
**while** Not converged **do**
    Sample a batch of tasks $\boldsymbol{b} \sim p^B(\tau)$.
    **for** All $\tau \in \boldsymbol{b}$ **do**
        **for** $k = 1, 2, \cdots, K$ **do**
            Sample $\kappa_k$ from $\pi_\kappa(\mathcal{D}^{\mathrm{tr}}_\tau \cup \mathcal{D}^{\mathrm{val}}_\tau)$;
            Get model representation $\boldsymbol{e}^{\kappa_k}_\tau = \psi(g(\kappa_k; \theta))$;
        **end for**
        Get model representation $\boldsymbol{e}^*_\tau = \psi(g(\mathcal{D}^{\mathrm{tr}}_\tau \cup \mathcal{D}^{\mathrm{val}}_\tau; \theta))$;
        Get inner-task distance $d^{\mathrm{in}}_\tau$ by (1);
        Get task-specific model $h_\tau = g(\mathcal{D}^{\mathrm{tr}}_\tau; \theta)$;
        Get validation loss $\mathcal{L}(\mathcal{D}^{\mathrm{val}}_\tau; h_\tau)$;
    **end for**
    Get $d^{\mathrm{in}} = \frac{1}{B}\sum_{\tau \in \boldsymbol{b}} d^{\mathrm{in}}_\tau$ and $d^{\mathrm{out}}$ by (2);
    Get loss $\mathcal{L}_{\mathrm{ConML}}$ by (3);
    Update $\theta$ by $\theta \leftarrow \theta - \nabla_\theta \mathcal{L}$.
**end while**

---

**Algorithm 4** ConML ($K = 1$).

---

**Input:** Task distribution $p(\tau)$, batch size $B$ (inner-task sample times $K = 1$ and sampling strategy $\pi_\kappa(\mathcal{D}^{\mathrm{tr}}_\tau \cup \mathcal{D}^{\mathrm{val}}_\tau) = \mathcal{D}^{\mathrm{tr}}_\tau$).
**while** Not converged **do**
    Sample a batch of tasks $\boldsymbol{b} \sim p^B(\tau)$.
    **for** All $\tau \in \boldsymbol{b}$ **do**
        Get task-specific model $h_\tau = g(\mathcal{D}^{\mathrm{tr}}_\tau; \theta)$, and model representation $\boldsymbol{e}^{\kappa_k}_\tau = \psi(g(\kappa_k; \theta))$;
        Get model representation $\boldsymbol{e}^*_\tau = \psi(g(\mathcal{D}^{\mathrm{tr}}_\tau \cup \mathcal{D}^{\mathrm{val}}_\tau; \theta))$;
        Get inner-task distance $d^{\mathrm{in}}_\tau$ by (1);
        Get validation loss $\mathcal{L}(\mathcal{D}^{\mathrm{val}}_\tau; h_\tau)$;
    **end for**
    Get $d^{\mathrm{in}} = \frac{1}{B}\sum_{\tau \in \boldsymbol{b}} d^{\mathrm{in}}_\tau$ and $d^{\mathrm{out}}$ by (2);
    Get loss $\mathcal{L}_{\mathrm{ConML}}$ by (3);
    Update $\theta$ by $\theta \leftarrow \theta - \nabla_\theta \mathcal{L}$.
**end while**

---

**Algorithm 5** ICL with ConML (ICL w/ ConML).

**Input:** Task distribution $p(\tau)$, batch size $B$, inner-task sample times $K$ and sampling strategy $\pi_\kappa$, dummy input $u$ (probe).
**while** Not converged **do**
    Sample a batch of tasks $\boldsymbol{b} \sim p^B(\tau)$.
    **for** All $\tau \in \boldsymbol{b}$ **do**
        **for** $k = 1, 2, \cdots, K$ **do**
            Sample $\kappa_k$ from $\pi_\kappa(\mathcal{D}_\tau)$;
            Get $\boldsymbol{e}_\tau^{\kappa_k} = g([\vec{\kappa_k}, u]; \theta)$;
        **end for**
        Get $\boldsymbol{e}_\tau^* = g([\vec{\mathcal{D}}_\tau, u]; \theta)$;
        Get inner-task distance $d_\tau^{\text{in}}$ by (1);
        Get task loss $\frac{1}{m} \sum_{i=0}^{m-1} \ell(y_{\tau,i+1}, g([\vec{\mathcal{D}}_{\tau,0:i}, x_{\tau,i+1}]; \theta))$;
    **end for**
    Get $d^{\text{in}} = \frac{1}{B} \sum_{\tau \in \boldsymbol{b}} d_\tau^{\text{in}}$ and $d^{\text{out}}$ by (2);
    Get episodic loss $\mathcal{L}_e = \frac{1}{B} \sum_{\tau \in \boldsymbol{b}} \frac{1}{m} \sum_{i=0}^{m-1} \ell(y_{\tau,i+1}, g([\vec{\mathcal{D}}_{\tau,0:i}, x_{\tau,i+1}]; \theta))$
    Update $\theta$ by $\theta \leftarrow \theta - \nabla_\theta(\mathcal{L}_e + \lambda(d^{\text{in}} - d^{\text{out}}))$.
**end while**

---

**Algorithm 6** MAML w/ ConML.

**Input:** Task distribution $p(\tau)$, batch size $B$, inner-task sample times $K = 1$ and sampling strategy $\pi_\kappa$
**while** Not converged **do**
    Sample a batch of tasks $\boldsymbol{b} \sim p^B(\tau)$.
    **for** All $\tau \in \boldsymbol{b}$ **do**
        **for** $k = 1, 2, \cdots, K$ **do**
            Sample $\kappa_k$ from $\pi_\kappa(\mathcal{D}_\tau^{\text{tr}} \cup \mathcal{D}_\tau^{\text{val}})$;
            Get model representation $\boldsymbol{e}_\tau^{\kappa_k} = \theta - \nabla_\theta \mathcal{L}(\kappa_k; h_\theta)$;
        **end for**
        Get model representation $\boldsymbol{e}_\tau^* = \theta - \nabla_\theta \mathcal{L}(\mathcal{D}_\tau^{\text{tr}} \cup \mathcal{D}_\tau^{\text{val}}; h_\theta)$.
        Get inner-task distance $d_\tau^{\text{in}}$ by (1);
        Get task-specific model $h_{\theta - \nabla_\theta \mathcal{L}(\mathcal{D}_\tau^{\text{tr}}; \theta)}$;
        Get validation loss $\mathcal{L}(\mathcal{D}_\tau^{\text{val}}; h_{\theta - \nabla_\theta \mathcal{L}(\mathcal{D}_\tau^{\text{tr}}; h_\theta)})$;
    **end for**
    Get $d^{\text{in}} = \frac{1}{B} \sum_{\tau \in \boldsymbol{b}} d_\tau^{\text{in}}$ and $d^{\text{out}}$ by (2);
    Get loss $\mathcal{L}_{\text{ConML}}$ by (3);
    Update $\theta$ by $\theta \leftarrow \theta - \nabla_\theta \mathcal{L}$.
**end while**

---

**Algorithm 7** Reptile w/ ConML.

**Input:** Task distribution $p(\tau)$, batch size $B$. (inner-task sample times $K = 1$ and sampling strategy $\pi_\kappa(\mathcal{D}_\tau^{\text{tr}} \cup \mathcal{D}_\tau^{\text{val}}) = \mathcal{D}_\tau^{\text{tr}}$)
**while** Not converged **do**
    Sample a batch of tasks $\boldsymbol{b} \sim p^B(\tau)$.
    **for** All $\tau \in \boldsymbol{b}$ **do**
        **for** $k = 1, 2, \cdots, K$ **do**
            Sample $\kappa_k$ from $\pi_\kappa(\mathcal{D}_\tau)$;
            Get model representation $\boldsymbol{e}_\tau^{\kappa_k} = \theta - \nabla_\theta \mathcal{L}(\kappa_k; h_\theta)$;
        **end for**
        Get model representation $\boldsymbol{e}_\tau^* = \theta - \nabla_\theta \mathcal{L}(\mathcal{D}_\tau^{\text{tr}} \cup \mathcal{D}_\tau^{\text{val}}; h_\theta)$.
        Get inner-task distance $d_\tau^{\text{in}}$ by (1);
    **end for**
    Get $d^{\text{in}} = \frac{1}{B} \sum_{\tau \in \boldsymbol{b}} d_\tau^{\text{in}}$ and $d^{\text{out}}$ by (2);
    Update $\theta$ by $\theta \leftarrow \theta + \frac{1}{B} \sum_{\tau \in \boldsymbol{b}} (\boldsymbol{e}_\tau^* - \theta) - \lambda \nabla_\theta (d^{\text{in}} - d^{\text{out}})$.
**end while**

---

**Algorithm 8** SCNAPs w/ ConML.

---

**Note:** Here $h_w$ corresponds to the feature extractor $f_\theta$; $H_\theta$ corresponds to the task encoder $g_\phi$ in [4].
**Input:** Task distribution $p(\tau)$, batch size $B$, inner-task sample times $K$ and sampling strategy $\pi_\kappa$.
Pretrain $h_w$ with the mixture of all meta-training data;
**while** Not converged **do**
    Sample a batch of tasks $\boldsymbol{b} \sim p^B(\tau)$.
    **for** All $\tau \in \boldsymbol{b}$ **do**
        **for** $k = 1, 2, \cdots, K$ **do**
            Sample $\kappa_k$ from $\pi_\kappa(\mathcal{D}_\tau^{\text{tr}} \cup \mathcal{D}_\tau^{\text{val}})$;
            Get model representation $\boldsymbol{e}_\tau^{\kappa_k} = H_\theta(\kappa_k)$;
        **end for**
        Get model representation $\boldsymbol{e}_\tau^* = H_\theta(\mathcal{D}_\tau^{\text{tr}} \cup \mathcal{D}_\tau^{\text{val}})$;
        Get inner-task distance $d_\tau^{\text{in}}$ by (1);
        Get task-specific model by FiLM $h_\tau = h_{w, H_\theta(\mathcal{D}_\tau^{\text{tr}})}$;
        Get validation loss $\mathcal{L}(\mathcal{D}_\tau^{\text{val}}; h_\tau)$;
    **end for**
    Get $d^{\text{in}} = \frac{1}{B} \sum_{\tau \in \boldsymbol{b}} d_\tau^{\text{in}}$ and $d^{\text{out}}$ by (2);
    Get loss $\mathcal{L}_{\text{ConML}}$ by (3);
    Update $\theta$ by $\theta \leftarrow \theta - \nabla_\theta \mathcal{L}$.
**end while**

---

---

**Algorithm 9** ProtoNet w/ ConML ($N$-way classification).

---

**Input:** Task distribution $p(\tau)$, batch size $B$, inner-task sample times $K = 1$ and sampling strategy $\pi_\kappa$
**while** Not converged **do**
    Sample a batch of tasks $\boldsymbol{b} \sim p^B(\tau)$.
    **for** All $\tau \in \boldsymbol{b}$ **do**
        **for** $k = 1, 2, \cdots, K$ **do**
            Sample $\kappa_k$ from $\pi_\kappa(\mathcal{D}_\tau^{\text{tr}} \cup \mathcal{D}_\tau^{\text{val}})$;
            Calculate prototypes $\boldsymbol{c}_j = \frac{1}{|\kappa_{k,j}|} \sum_{(x_i,y_i) \in \kappa_{k,j}} f_\theta(x_i)$ for $j = 1, \cdots, N$;
            Get model representation $\boldsymbol{e}_\tau^{\kappa_k} = [\boldsymbol{c}_1 | \boldsymbol{c}_2 | \cdots | \boldsymbol{c}_N]$;
        **end for**
        Calculate prototypes $\boldsymbol{c}_j = \frac{1}{|\mathcal{D}_j|} \sum_{(x_i,y_i) \in \mathcal{D}_j} f_\theta(x_i)$ for $j = 1, \cdots, N$;
        Get model representation $\boldsymbol{e}_\tau^* = [\boldsymbol{c}_1 | \boldsymbol{c}_2 | \cdots | \boldsymbol{c}_N]$;
        Get inner-task distance $d_\tau^{\text{in}}$ by (1);
        Get task-specific model $h_{[\boldsymbol{c}_1|\boldsymbol{c}_2|\cdots|\boldsymbol{c}_N]}$, which gives prediction by $p(y = j \mid x) = \frac{exp(-d(f_\theta(x), \boldsymbol{c}_j))}{\sum_{j'} exp(-d(f_\theta(x), \boldsymbol{c}_{j'}))}$;
        Get validation loss $\mathcal{L}(\mathcal{D}_\tau^{\text{val}}; h_{[\boldsymbol{c}_1|\boldsymbol{c}_2|\cdots|\boldsymbol{c}_N]})$;
    **end for**
    Get $d^{\text{in}} = \frac{1}{B} \sum_{\tau \in \boldsymbol{b}} d_\tau^{\text{in}}$ and $d^{\text{out}}$ by (2);
    Get loss $\mathcal{L}_{\text{ConML}}$ by (3);
    Update $\theta$ by $\theta \leftarrow \theta - \nabla_\theta \mathcal{L}$.
**end while**

---

# D ICL with ConML

## D.1 ICL

ICL is first proposed for LLMs [6], where examples in a task are integrated into the prompt (input-output pairs) and given a new query input, the language model can generate the corresponding output. This approach allows pretrained model to address new tasks without fine-tuning the model. For example, given "*happy->positive; sad->negative; blue->*", the model can output "*negative*", while given "*green->cool; yellow->warm; blue->*" the model can output "*cool*". ICL has the ability to learn from the prompt. Training ICL can be viewed as learning to learn, i.e., meta-learning [29, 16, 24]. More generally, the input and output are not necessarily to be natural language. In ICL, a sequence model $T_\theta$ (typically transformer [45]) is trained to map sequence $[x_1, y_1, x_2, y_2, \cdots, x_{m-1}, y_{m-1}, x_m]$ (prompt prefix) to prediction $y_m$. Given distribution $P$ of training prompt $t$, then training ICL follows an auto-regressive manner:

$$\min_\theta \mathbb{E}_{t \sim P(t)} \frac{1}{m} \sum_{i=0}^{m-1} \ell(y_{t,i+1}, T_\theta([x_{t,1}, y_{t,1}, \cdots, x_{t,i+1}])). \tag{6}$$

It has been mentioned that the training of ICL can be viewed as an instance of meta-learning [16, 1] as $T_\theta$ learns to learn from prompt. It has been pointed out that ICL model is meta-learner with minimal inductive bias [53]. In this section we first formally reformulate $T_\theta$ to meta-learner $g(; \theta)$, then introduce how ConML can be integrated with ICL.

## D.2 A Meta-learning Reformulation

Denote a sequentialized $\mathcal{D}$ as $\vec{\mathcal{D}}$ where the sequentializer is default to bridge $p(\tau)$ and $P(t)$. Then the prompt $[x_{\tau,1}, y_{\tau,1}, \cdots, x_{\tau,m}, y_{\tau,m}]$ can be viewed as $\vec{\mathcal{D}}_\tau^{tr}$ which is providing task-specific information. Note that ICL does not specify an explicit output model $h(x) = g(\mathcal{D}; \theta)(x)$; instead, this procedure exists only implicitly through the feeding-forward of the sequence model, i.e., task-specific prediction is given by $g([\vec{\mathcal{D}}, x]; \theta)$. Thus we can reformulate the training of ICL (6) as:

$$\min_\theta \mathbb{E}_{\tau \sim p(\tau)} \frac{1}{m} \sum_{i=0}^{m-1} \ell(y_{\tau,i+1}, g([\vec{\mathcal{D}}_{\tau,0:i}, x_{\tau,i+1}]; \theta)). \tag{7}$$

The loss in (7) can be evaluated through episodic meta-training, where each task in each episode is sampled multiple times to form $\mathcal{D}_\tau^{val}$ and $\mathcal{D}_\tau^{tr}$ to evaluate the episodic loss $\mathcal{L}_e$ in an auto-regressive manner. The training of ICL thus follows the episodic meta-training (Algorithm 1), where the validation loss with determined $\mathcal{D}_\tau^{tr}$ and $\mathcal{D}_\tau^{val}$: $\mathcal{L}(\mathcal{D}_\tau^{val}; g(\mathcal{D}_\tau^{tr}; \theta))$, is replaced by loss validated in the auto-regressive manner: $\frac{1}{m} \sum_{i=0}^{m-1} \ell(y_{\tau,i+1}, g([\vec{\mathcal{D}}_{\tau,0:i}, x_{\tau,i+1}]; \theta))$.

## D.3 Integrating ConML with ICL

Since the training of ICL could be reformulated as episodic meta-training, the three steps to measure ConML proposed in Section 3 can be also adopted for ICL, but the first step to obtain model representation $\psi(g(\mathcal{D}, \theta))$ needs modification. Due to the absence of an inner learning procedure for a predictive model for prediction $h(x) = g(\mathcal{D}; \theta)(x)$, representation by explicit model weights of $h$ is not feasible for ICL.

To represent what $g$ learns from $\mathcal{D}$, we design to incorporate $\vec{\mathcal{D}}$ with a dummy input $u$, which functions as a probe and its corresponding output can be readout as representation:

$$\psi(g(\mathcal{D}; \theta)) = g([\vec{\mathcal{D}}, u]; \theta), \tag{8}$$

where $u$ is constrained to be in the same shape as $x$, and has consistent value in an episode. The complete algorithm of ConML for ICL is in Appendix C.For example, for training a ICL model on linear regression tasks we can choose $u = \mathbf{1}$, and in pretraining of LLM we can choose $u =$"*what is this task?*". From the perspective of learning to learn, ConML encourages ICL to align and discriminate like it does for conventional meta-learning, while the representations to evaluate inner- and inter- task distance are obtained by probing output rather than explicit model weights. Thus, incorporating ConML into the training process of ICL benefits the fast-adaptation and task-level generalization ability. From the perspective of supervised learning, ConML is performing unsupervised data augmentation that it introduces the dummy input and contrastive objective as additional supervision to train ICL.

# E    Experimental Results on Synthetic Data

We begin by conducting experiments on synthetic data in a controlled setting to explain: (i) Does ConML enable meta-learners to develop alignment and discrimination abilities? (ii) How do alignment and discrimination boost meta-learning performance? We take MAML w/ ConML as example and investigate above questions with few-shot regression problem following the same settings in [15]. Each task involves regressing from the input to the output of a sine wave, where the amplitude and phase of the sinusoid are varied between tasks. The amplitude varies within $[0.1, 5.0]$ and the phase varies within $[0, \pi]$. This synthetic regression dataset allows us to sample data and adjust the distribution as necessary for analysis. The implementation of ConML follows a simple intuitive setting: inner-task sampling $K = 1$ and $\pi_\kappa(\mathcal{D}_\tau^{\text{tr}} \cup \mathcal{D}_\tau^{\text{val}}) = \mathcal{D}_\tau^{\text{tr}}$, $\phi(a, b) = 1 - {a \cdot b}/{\|a\|\|b\|}$ (cosine distance) and $\lambda = 0.1$. The meta-leaner is trained on meta-training distribution with amplitudes uniformly distributed over $[0.1, 5]$, and each training task has a fixed $N = 10$. For Figure 4(f), the meta-learner is tested on tasks with amplitudes uniformly distributed over $[0.1 + \delta, 5 + \delta]$, where $\delta$ is shown on the $x$-axis.

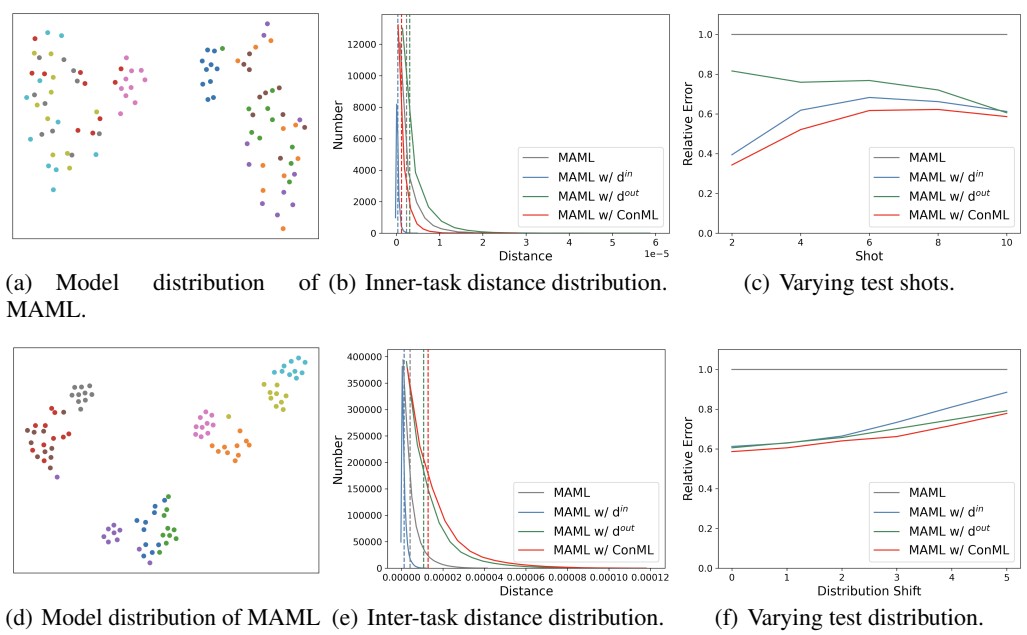

(a) Model distribution of MAML.

(b) Inner-task distance distribution.

(c) Varying test shots.

(d) Model distribution of MAML w/ ConML.

(e) Inter-task distance distribution.

(f) Varying test distribution.

Figure 4: Evaluation of ConML on synthetic few-shot regression.

**Learning to learn with ConML brings generalizable alignment and discrimination abilities.** If optimizing $d^{\text{in}}$ and $d^{\text{out}}$ does equip meta-learner with generalizable alignment and discrimination, MAML w/ ConML can generate more similar models from different subsets of the same task, while generating more separable models from different tasks, though these tasks are unseen. This can be verified by evaluating the clustering performance for model representations $e$ from unseen tasks. Figure 4(a) and 4(d) visualize the distribution of these models, where each point corresponds to the result of a subset and the same color indicates sampled from the same task. We randomly sample 10 different unseen tasks. For each task, we sample 10 different subsets, each containing $N = 10$ samples. Using these 100 different training sets $\mathcal{D}^{\text{tr}}$ as input, the meta-learner generates 100 models. It can be obviously observed MAML w/ ConML performs better alignment and discrimination than MAML. To quantity the results, we also evaluate the supervised clustering performance, where task identity is used as label. Table 6 shows the supervised clustering performance of different metrics: Silhouette score [37] and Calinski-Harabasz index (CHI) [7]. The results indicate that MAML with ConML significantly outperforms standard MAML across all metrics. These findings confirm that training with ConML enables meta-learners to develop alignment and discrimination abilities that generalize to meta-testing tasks.

Table 6: Meta-testing performance (MSE) on few-shot regression problem and clustering performance (Silhouette and CHI) of model representations.

| Method | MSE (5-shot) | MSE (10-shot) | Silhouette | CHI |
|---|---|---|---|---|
| MAML | $.677 \pm .038$ | $.068 \pm .002$ | $.107 \pm .060$ | $31.6 \pm 2.5$ |
| MAML w/ ConML | $\mathbf{.394} \pm .010$ | $\mathbf{.040} \pm .001$ | $\mathbf{.195} \pm .062$ | $\mathbf{39.2} \pm 2.6$ |

**Alignment enhances fast-adaptation and discrimination enhances task-level generalizability.** We aim to understand the individual contributions of optimizing $d^{\text{in}}$ (alignment) and $d^{\text{out}}$ (discrimination) to meta-learning performance. In conventional unsupervised contrastive learning, both positive and negative pairs are necessary to avoid learning representations without useful information. However, in ConML, the episodic loss $\mathcal{L}_e$ plays a fundamental role in "learning to learn," while the contrastive objective serves as additional supervision to enhance alignment and discrimination. Thus, we consider two variants of ConML: **MAML w/ $d^{\text{in}}$** which optimizes $\mathcal{L}_e$ and $d^{\text{in}}$, **MAML w/ $d^{\text{out}}$** which optimizes $\mathcal{L}_e$ and $d^{\text{out}}$. Figure 4(b) and 4(e) visualize the distribution of $d^{\text{in}}$ and $d^{\text{out}}$ respectively, where the dashed lines mark mean values. We randomly sample 1000 different unseen tasks, with 10 different subsets (each containing $N = 10$ samples) per task. These subsets are aggregated into a single set of $N = 100$ to obtain $e_\tau^*$ for each task. Smaller $d^{\text{in}}$ means better alignment and larger $d^{\text{out}}$ means better discrimination. We can find that the alignment and discrimination abilities are separable, generalizable, and that ConML effectively couples both. Figure 4(c) shows the testing performance given different numbers of examples per task (shot). The results indicate that the improvement from alignment (MAML w/ $d^{\text{in}}$) is more pronounced in few-shot scenarios, highlighting its close relationship with fast-adaptation. Figure 4(f) shows the out-of-distribution testing performance. As the distribution gap increases, the improvement from discrimination (MAML w/ $d^{\text{out}}$) is more significant than from alignment (MAML w/ $d^{\text{in}}$), indicating that discrimination plays a critical role in task-level generalization. ConML leverages the benefits of both alignment and discrimination.

# F    Experimental Results Obtained Using Different Backbones

In the main text, we have used the following backbones in experiment: **Conv4**: MAML, FOMAML, Reptile, MatchNet, ProtoNet; **ResNet12**: MELR, Lastshot; **ResNet18**: SCNAPs; **ViT-base**: CAML.

To study the effect of ConML on different backbones, we compare MAML w/o and w/ ConML, reporting the miniImageNet 5-way 1-shot accuracy. We specifically demonstrate the effect of equipping the model with ConML using the change in accuracy ($\Delta$ Acc).

Table 7: Meta-testing accuracy (%) on miniImageNet 5-way 1-shot, using different backbones.

| Backbone | Conv4 | Conv6 | ResNet12 | ResNet18 |
|---|---|---|---|---|
| MAML w/o ConML | 48.7 | 50.9 | 57.2 | 56.3 |
| MAML w/ ConML | 56.2 | 57.8 | 64.5 | 64.9 |
| $\Delta$ Acc | +7.5 | +6.9 | +7.3 | +8.6 |

Table 7 shows the results. As the network deepens, $\Delta$ Acc shows little change. Although deeper networks generally achieve higher baseline accuracy, making further improvements challenging, ConML consistently enhances performance—even outperforming shallower architectures. For instance, while ResNet18 (which may be overly deep for 1-shot MAML on miniImageNet) generalizes worse than the shallower ResNet12 without ConML, ConML boosts ResNet18's by a significant $\Delta$ Acc (+8.6%), surpassing ResNet12 with ConML. This suggests that deeper networks can better leverage ConML's alignment and discrimination capabilities.

# G Implementation Details

## G.1 Model Analysis

ConML optimizes the following objective: $\mathcal{L}_{\text{ConML}} = \mathcal{L}_e + \lambda\mathcal{L}_c$, where $\mathcal{L}_e$ is the episodic loss, and $\mathcal{L}_c$ is the contrastive loss. In the previous sections, to highlight our motivation and perform a decoupled analysis, we used the naive contrastive loss $\mathcal{L}_c = d^{\text{in}} - d^{\text{out}}$, with the natural cosine distance $\phi(x, y) = 1 - \frac{x^\top y}{\|x\|\|y\|}$. Here, we also considered a manually bounded Euclidean distance $\phi(x, y) = \text{sigmoid}(\|x - y\|)$. Beyond the simple contrastive loss, we incorporate the InfoNCE [32] loss for an episode with a batch $b$ containing $B$ tasks. The contrastive loss is defined as $\mathcal{L}_c = -\sum_{\tau \in b} \log \left( \frac{\exp(-D_\tau^{in})}{\exp(-D_\tau^{in}) + \sum_{\tau' \in b\backslash\tau} \exp(-D_{\tau,\tau'}^{out})} \right)$, where $D_{\tau,\tau'}^{out} = \phi(e_\tau^*, e_{\tau'}^*)$. In this case, we treat negative "distance" as "similarity." For the similarity metric in InfoNCE, we experiment with both cosine distance $\phi(x, y) = 1 - \frac{x^\top y}{\|x\|\|y\|}$ and Euclidean distance $\phi(x, y) = \|x - y\|$.

## G.2 ICL

We implement ICL w/ ConML with $K = 1$ and $\pi_\kappa([x_1, y_1, \cdots, x_n, y_n]) = [x_1, y_1, \cdots, x_{\lfloor\frac{n}{2}\rfloor}, y_{\lfloor\frac{n}{2}\rfloor}]$. To obtain the implicit representation (4), we sample $u$ from a standard normal distribution (the same with $x$'s distribution) independently in each episode. Since the output of (4) is a scalar, i.e., representation $e \in \mathbb{R}$, we adopt distance measure $\phi(a, b) = \sigma((a - b)^2)$, where $\sigma(\cdot)$ is sigmoid function to bound the squared error. $\lambda = 0.02$. Note that the learning of different functions (LR, DT, SLR, NN) share the same efficient and straightforward settings about ConML above, which shows ConML can bring ICL universal improvement with cheap implementation.

We notice that during training of LR and SLR $\lfloor\frac{n}{2}\rfloor = 5$, which happens to equals to the dimension of the regression task. This means sampling by $\pi_\kappa$ would results in the minimal sufficient information to learn the task. In this case, minimizing $d^{\text{in}}$ is particularly beneficial for the fast-adaptation ability, shown as Figure 3(a) and 3(b). This indicates that introducing prior knowledge to design the hyperparameter settings of ConML could bring more advantage.

