# OpenReview forum: "Learning to Learn with Contrastive Meta-Objective"
_NeurIPS.cc/2025/Conference — NeurIPS 2025 oral_

### Official Review · Reviewer_Kgc2 · 2025-06-27

**Clarity:** 3
**Significance:** 4
**Originality:** 3
**Rating:** 4
**Confidence:** 4

**Summary:**

This paper proposes a general meta-learning framework called ConML (Contrastive Meta-Learning), which aims to improve the generalization performance of meta-learners by simulating the "alignment" and "discrimination" abilities of human learning. The core idea is to introduce an additional "contrastive meta-objective" in the training phase of meta-learning. Specifically, ConML performs contrastive learning in the "model space": on the one hand, it achieves "alignment" by minimizing the distance between models learned from different data subsets under the same task; on the other hand, it achieves "discrimination" by maximizing the distance between models learned from different tasks. The author demonstrates that ConML is a problem- and learner-agnostic framework that can be seamlessly integrated into a variety of existing meta-learning algorithms (such as optimization-based, metric-based, and amortization-based methods) as well as the emerging In-Context Learning (ICL).

**Questions:**

1. Regarding the model representation of ICL (Section 3.3): Can you elaborate on the criteria for choosing the "virtual input" $u$? Have you tried different $u$, and how does its choice affect the final performance? Are there any specific design considerations behind keeping $u$ consistent within an episode but sampled independently across episodes?

2. Regarding the theoretical analysis (Section 3.2): The proof outline in Appendix B relies on the assumption that minimizing $\mathcal{L}_c$ makes the distribution of model representations $P_g(\theta)$ symmetric. Can you provide more intuition on why this assumption is reasonable for sufficiently expressive models and how it relates to practical implementations of contrastive loss?

3. In cross-domain experiments (Table 3), ConML shows very little to no performance improvement on some datasets (e.g., fo-Proto-MAML on Birds). Is there a potential explanation for why ConML works better in some domains than others? Does it have to do with the diversity or similarity of tasks in these datasets?

**Ethical Concerns:**

["NO or VERY MINOR ethics concerns only"]

**Final Justification:**

The rebuttal has addressed my question to some extent. However, my rating remains the same primarily due to the concern about memory cost: 4: Borderline accept.

**Limitations:**

Yes. The authors explain the limitations of their study, and their assessment of potential societal impacts meets the standards for basic research of this type.

**Paper Formatting Concerns:**

There are no major formatting issues.

**Quality:**

2

**Strengths And Weaknesses:**

### Strengths
1. Extending the idea of ​​contrastive learning from the traditional "feature space" to the "model space" of meta-learning is a novel and insightful idea.
2. The highlight of ConML is its versatility. The authors clearly show how to apply the framework to different categories of meta-learning algorithms and provide specific implementation specifications in Section 3.3 and Table 1.
3. The paper provides a theoretical perspective in Section 3.2, proving that minimizing the contrast loss helps to reduce an upper bound on the generalization error.
4. The paper is well-written and logically clear.


### Weaknesses
1. The paper claims that ConML only brings a "small additional computational cost." However, as seen in Table 4, after enabling ConML (K=1), the memory usage increases significantly (for example, MAML increases from 1331MB to 2801MB, more than 2 times). Although the relative time increase is not significant (about 1.1 times), the jump in memory is not negligible, especially when the model is larger or the K value is higher.
2. The model analysis in Figure 2 shows that the performance is quite sensitive to the choice of $\lambda$, distance function (Euclidean/cosine), and loss form (InfoNCE/naive form). Therefore, the statement that "not much tuning effort is required" may be overly optimistic.
3. The approach to defining a model representation for ICL using a "dummy input" $u$. However, its theoretical foundation is somewhat weak. How sensitive is the final performance to the choice of $u$? The paper gives examples of $u$=1 or $u$= "what is this task?" but there is no in-depth analysis or justification for this choice.
4. If there is overlap or even similarity between tasks, it is not reasonable to directly regard them as negative samples. For example, task a (class 1 2 3) and task b (class 1 3 4).
5. Learning-to-learn is very general. As usual, learning-to-learn methods are also validated on reinforcement learning.

---

> ### Author Rebuttal · Authors · 2025-07-31
>
> Thanks a lot for your valuable comments and suggestions. We find most of them very objective, indicating the reviewer's accurate understanding.
>
> ## W1. Memory consumption
> Indeed, ConML introduces additional memory consumption which grows about linearly with K. We want to emphasize that $K=1$ is good enough all our experiment, and the memory consumption (about 2 times) is at the same scale with w/o ConML.
>
> ## W2. Hyperparameter sensitivity
> Our statement "not much tuning effort is required" is because all main results (Table 2 & 3) are obtained with the shared and primitive hyperparameters (lin 254), not even the best results we have witnessed (only corresponding to the red line in Figure 2). And in a large range of choice of hyperparameter, ConML brings improvement over baseline (gray line in Figure 2).
>
> ## W3. $u$ for ICL with ConML
> Please refer to Q1.
>
> ## W4. Task distribution Problem
> we think it is a very rational point and were taking it as future directions.
> As we have discussed in the manuscript, we mainly intend to provide the idea and investigate the role of contrastive meta-objective, and there are still many directions to further improve the performance, e.g., contrastive function, distance metric, inner-(subset) and inter-(batch)task sampling strategy.
>
> The mentioned problem involves the inter-(batch)task sampling strategy we have discussed. Optimizing inter-task distance with the current random task batch sampling can effect sub-optimally in scenarios pointed out by the reviewer. It just works statistically.
> This might be improved by advanced task batch sampling strategy with task-level diversity/difficulty/similarity awareness and schedule, which have been studied in literature, e.g., [1,2,3].
>
> [1]  Adaptive task sampling for meta-learning. ECCV2020
>
> [2] The effect of diversity in meta-learning. AAAI2023
>
> [3] Towards task sampler learning for meta-learning. IJCV2024
>
> ## W5. Broader scope in Meta-learning
> Thanks for your kind advice. We will try to apply ConML on other meta-learning scenarios including RL as future work.
>
> ## Q1. Choosing the "virtual input" $u$ for ICL.
> First, we want to illustrate why $u=1$ meets ConML's goal of additional supervision by task identity without labeled data for generalization.
> Let's consider ICL for regression tasks $y=wx$, where the output of Eqn(4) is expected to be $wu$. ConML expect same $w$ can be learned from different subsets of the same task, and distinguishable $w$ can be learned from different tasks. However, $w$ is intractable in ICL models, so ConML use the output of Eqn(4) as an alternative, which is exactly $w$ with $u=1$. Note that above goal only requires $u$ to be constant, where $u=1$ is only an example and choosing $u$ to be any constant is feasible.
>
> Now let's discuss our consideration of keeping $u$ consistent within an episode but sampled independently across episodes, while keeping $u$ consistent globally (e.g. $u=1$) already meets the goal of ConML. Note that in ConML algorithm, the procedure of calculating and optimizing inner- and inter-task distance is complete in each episode, independent to other episodes. So there is no constraint on $u$ to be consistent across episodes. From the principle of "Occam's Razor", we set $u$ free across episodes by sampling independently in each episode. From another perspective, technically we can not define a optimal constant as $u$, and keeping $u$ globally consistent might result in model collapsing with large $\lambda$.
>
> Empircally, we have evaluated setting $u=1$ under the same metric as Table 5. We provide results in the following table (the relative gain of $u=1$ over independent $u$ across episodes is in bracket). We find that though not significant, such independent design makes general improvement.
>
> Function | LR | DT
> -|-|-
> Rel. Min. Error  | 0.35 (-17%) |0.62 (-23%)
> Shot Spare  | -3.86 (-18%) | -3.80 (-10%)
>
> ## Q2. Why minimizing $L_c$ makes $P_g$ symmetric
> Thanks for your detailed reading! There might be a little misunderstanding.
> Technically, it is not an "assumption" but a solution to an optimization problem, which can be formally described as $argmin_{p(\tau),p(x|\tau)} E_{i,j,k \sim p^3(\tau), x_\tau\sim p(x|\tau)}[\frac{x_{i,1}^\top x_{i,2}}{||x_{i,1}||||x_{i,2}||}-\frac{x_j^\top x_k}{||x_j||||x_k||}]$. And it is easy to see the solution is $\forall p(\tau), s.t. p(\tau)=p(-\tau), p(x|\tau) =\delta(\tau)$. Then in the manuscript $P(X)=p(x|\tau)p(\tau)$ is symmetric.
>
> ## Q3. Explanation of performance on Meta-Dataset.
> We understand this concern may rise from similar reasons with weakness 4. We agree with such potential problem as discussed in "A reply to Weakness". But we think it does not holds here considering the setting of Meta-Dataset. Following the common setting of Meta-Dataset, we collect all meta-training tasks in ILSVRC, Omniglot, Aircraft, Birds, Textures, Quick Draw, Fungi, VGG Flower datasets for meta-training, then test on meta-testing tasks from both aforementioned datasets (in-domain) and additional Traffic Signs and MS COCO datasets (ood) to perform cross-domain meta-learning. That is to say, we are using the same commonly meta-trained learner when testing on different datasets. So the mentioned reason could not explain well.
>
> We apologize for currently we could not explain the failure of ConML on fo-Proto-MAML testing on Birds. We will analyze this further to see if this keeps occurring with different hyperparameters and vision models if necessary.

---

> > ### Comment · Reviewer_Kgc2 · 2025-08-04
> >
> > Dear Authors,
> >
> >   Thanks for your rebuttal. Your responses have clarified some of my questions. However, my concern regarding the method's robustness, especially in cross-domain settings, has not been fully resolved. I appreciate your honesty in acknowledging that you currently cannot explain why ConML failed to show improvement for fo-Proto-MAML on the "Birds" dataset. While I understand this may require further investigation, this unexplained failure case raises questions about the method's stability and the true extent of its claimed universality. Since being "problem- and learner-agnostic" is a cornerstone of your contribution, this particular result weakens that claim and suggests there are underlying conditions for your method's success that are not yet understood.  Furthermore, regarding the computational cost, I find the argument that a memory increase of over 100% constitutes a "small additional cost" to be unconvincing.
> >
> >   Given that these key concerns about the method's robustness and practical costs remain, I will maintain my original "Borderline Accept" rating. The work is innovative, but its limitations need to be more thoroughly addressed and understood.
> >
> > Best regards,
> > Reviewer Kgc2

---

> > > ### Author Response · Authors · 2025-08-05
> > > **Thanks for your response**
> > >
> > > We sincerely thank the reviewers for the detailed response and clarity of remaining concerns. Here we would like to provide more results about ConML's improvement for fo-Proto-MAML. Following the same setting as Table 3, we re-run fo-Proto-MAML with ConML.  We implement three groups of random seeds/hyperparmeters with slight modification from the reported one from Table 3. A: We change random seeds from 1,2,3 to 4,5,6; B: We modify $\lambda$ from 0.1 to 0.3; C: We modify $d^{in}-d^{out}$ to InfoNCE; The following results show that ConML with all configs improves fo-Proto-MAML generally, and can obtain better performance than reported by Table 3 if tuning hyperparameters. Thus we assume the issue of "ConML failed to show improvement for fo-Proto-MAML on the 'Birds' dataset" is a very special case caused by certain unique combination of configrations. The true effect of ConML has no lack of universality.
> > >
> > > Acc (\%) | w/o ConML | w/ ConML (A)| w/ ConML (B)| w/ ConML (C)
> > > -|-|-|-|-
> > > ILSVRC  | 49.5 |54.5| 54.6| 55.2
> > > Omniglot  | 63.3 | 70.4| 70.2| 71.1
> > > Aircraft |55.9 | 61.0| 62.2| 62.3
> > > Birds | 68.6| 69.5|72.5| 73.8
> > > Textures | 66.4 | 69.6|69.2| 70.7
> > > Quick Draw | 51.5| 53.5|54.0|54.5
> > > Fungi | 39.9 | 42.9| 44.3| 44.4
> > > VGG Flower | 87.1|90.4|90.8| 91.3
> > > Traffic Signs | 48.8|51.5| 52.7| 52.7
> > > MS COCO | 43.7|48.0|48.8| 49.5

---

> ### Comment · Reviewer_Kgc2 · 2025-08-05
>
> Dear Authors,
>
>   Thanks for your reply. It has addressed my question to some extent. However, my rating remains the same primarily due to the concern about memory cost.
>
> Best regards,
> Reviewer Kgc2

---

### Official Review · Reviewer_PV71 · 2025-06-28

**Clarity:** 3
**Significance:** 2
**Originality:** 2
**Rating:** 5
**Confidence:** 3

**Summary:**

The paper 'Learning to Learn with Contrastive Meta-Objective' introduces a contrastive loss in the model representation space. This loss is used to minimize the intra-task distances while maximize the inter-task distances. The goal of this loss is for better task identification during meta training. The contrastive loss can be straightforwardly applied to different paradigms meta-learning methods including gradient based, metric based, amortized and ICL. Numerical experiments seem to suggest promising improvements over the baseline when combining various meta-learning algorithm with this contrastive loss, with non-significant computation overhead.

**Questions:**

1. line 198 and line 200: why is $\theta$ only updated by ONE gradient step? In general, MAML optimize its inner problem by multiple steps sgd.
2. line 196-line 231: it would add to the clarity if the authors could write clearly whether $\mathcal{D}$ is the training set or validation set or both.
3. line 664-line 668 is repeated.

**Ethical Concerns:**

["NO or VERY MINOR ethics concerns only"]

**Final Justification:**

All my questions and concerns have been addressed by the authors. I will keep my positive rating.

**Limitations:**

Yes

**Quality:**

3

**Strengths And Weaknesses:**

Strengths:
1. This paper is well-written and well-organized. Notations and definitions are clear, which makes the paper easy to read. The proposed contrastive loss makes intuitive sense.
2. The contrastive loss can be directly applied to different types of meta-learning methods including optimization-based, metric-based, amortization-based and ICL, almost in a plug-and-play manner. This loss function is also problem-agnostic. Previous works are either problem-specific or learner-specific.
3. The authors have performed many numerical experiments with promising results and similar running time as the vanilla methods. Ablation studies on $\lambda$ and $K$ are also provided.

Weaknesses:
1. The main contribution of the work is proposing a the contrastive loss $\mathcal{L}_c$, which in most experiments takes the form $d^{in} - d^{out}$. To better understand this loss, the authors have explored Euclidean distance $\phi$ and $\mathcal{L}_c$ in the form of InfoNCE. Based on results from Figure 2, it seems that for a vanilla $\mathcal{L}_c$, Euclidean distance is more stable than cosine similarity for large $\lambda$. Could the authors provide more intuition on this phenomenon?
2. Also based on Figure 2, $\mathcal{L}_c$ based on InfoNCE consistently beats vanilla $\mathcal{L}_c$. Can this observation be generalized to other experiments in this work?

---

> ### Author Rebuttal · Authors · 2025-07-31
>
> Thanks for you appreciation and valuable comments. We especially appreciate your detailed reading.
> ## Q1. Distance metric and contrastive loss
> We would like to discuss the distance metric and contrastive function respectively.
>
> For the choice of distance metric, we believe that the matching between the chosen distance metric and model representation is the key to further success. The model representation is a set of model parameters,
> whose role in the meta-learner need to be considered.
> For example, we can find that Euc distance performs much better than cosine in ProtoNet (Fig.3),
> since ProtoNet makes classification with Euclidean distance, and ConML contrasts the classifier's weights — describing the model's behavior more precisely.
> Although cosine works generally, it would be interesting to tailor distance metrics for various parameter types (e.g., classifiers, MLPs, CNNs, GNNs, Transformers).
>
> For the contrastive function,
> InfoNCE generally obtains better performance than the naive $d^{in}-d^{out}$ (Figure 2), which we assume to be generalizable.
> While InfoNCE exploits same information as the naive loss does.
> The performance difference might be explained with the difference of gradient magnitudes, where InfoNCE matches the term of lower bound of mutual information between model representations.
> In most parts of this paper we use the naive $d^{in}-d^{out}$, to illustrate ConML more straightforwardly, and to decouple $d^{in}$ $d^{out}$ for detail investigation (Appendix E). As we have discussed in the manuscript, we mainly intend to provide the idea and investigate the role of contrastive meta-objective, and there are still many directions to further improve the performance, e.g., contrastive function, distance metric, inner-(subset) and inter-(batch)task sampling strategy.
>
> ## Q2. inner-update steps in MAML in line 198 and 200
> We acknowledge the concern appreciate the reviewer's carefulness.
> In line 198 and 200, we use $\theta-\nabla_\theta L$ just to show that the function of MAML as a meta-learner, is achieved by gradient descents. We only explicitly write one step for notation simplicity. ConML's framework allows, and our experiments in fact do, multiple steps inner-update for optimization-based meta-learners like MAML.
> Such simplified notation can be found in a lot of related papers, even in "Model-Agnostic Meta-Learning for Fast Adaptation of Deep Networks" (MAML) itself (sec 2.2).
>
> ## Q3. $D$'s identity in line 196-231
> Thanks for your kind suggestion. The $D$ in sec 3.3 can be various. For standard mini-batch episodic training (Algorithm 1), all $D$ in this part is $D^{tr}$. For ConML, considering the Algorithm 2 we introduced in sec 3.1, $D$ can be $D^{tr}$, $D^{tr}\cup D^{val}$ or $\kappa_k$.
> Section 3.3 is intended for specifying the general framework of ConML in sec 3.1 to different meta-learners. We hope readers consider this part based on above information, as a modularized part.
>
> ## Q4. Appendix line 664-line 668 is repeated.
> We sincerely apologize for the error and thank you again for your detailed reading. We will modify it in manuscript.

---

> > ### Comment · Reviewer_PV71 · 2025-08-05
> >
> > Thank you for the responses. My questions/concerns have been addressed by the authors. I will keep my current score.

---

### Official Review · Reviewer_euqW · 2025-07-02

**Clarity:** 2
**Significance:** 2
**Originality:** 3
**Rating:** 4
**Confidence:** 4

**Summary:**

This paper focuses on enhancing the alignment and discrimination capabilities of meta learners via a task-level contrastive meta-objective. Unlike traditional contrastive learning in representation space, the proposed method applies contrastive learning directly in the model space, which is a novel and insightful idea. Empirical studies show consistent performance improvements across multiple meta-learning algorithms and settings, which validates the effectiveness of the proposed ConML.

**Questions:**

- In Table 1, the model representations for optimization-based methods are high-dimensional model weights, which could make similarity calculations computationally intensive. It would be helpful if the authors could comment on the resulting computational overhead and whether it poses practical limitations.
- Additionally, there seems to be a discrepancy between the description of the meta learners provided in line 198 and the information presented in Table 1. Clarification regarding this inconsistency would improve the clarity of the paper.
- As described in line 204, metric-based algorithms are naturally suited for classification tasks, yet their performance reported in experiments is lower than other categories. What might explain this gap?

**Ethical Concerns:**

["NO or VERY MINOR ethics concerns only"]

**Final Justification:**

The responses addressed my concerns, and I will raise my score from 3 to 4.

**Limitations:**

- The paper does not explicitly clarify the relationship between in-context learning and meta-learning, which could be elaborated for better conceptual clarity.
- More references should be considered:

[1] Robust Fast Adaptation from Adversarially Explicit Task Distribution Generation. KDD2025.

[2] Data augmentation for meta-learning. ICML2021.

**Paper Formatting Concerns:**

No concerns.

**Quality:**

2

**Strengths And Weaknesses:**

**Strengths:**
- Universal and agnostic to both problem types and meta-learner architectures.
- Successfully applies to in-context learning, extending its impact beyond conventional meta learning.
- Provides theoretical guarantees showing tighter generalization error bounds when optimizing the contrastive objective.

**Weaknesses:**
- Some expressions are overly casual and could be made more precise or formal (e.g., “people also…” in line 27; “obtaining …” in lines 123, 130,139).
- Lacks evaluation of robustness under noisy or corrupted data, which is important for meta-learning generalization.
- Episodic learning inherently provides task-identity information. However, since tasks lie in a distribution, when task differences are subtle or overlapping, could ConML degrade performance or overfit?

---

> ### Author Rebuttal · Authors · 2025-07-31
>
> Thanks a lot for your valuable comments and suggestions.
>
> ## Q1. Lack of evaluation of robustness under noisy or corrupted data
> This is a very good point, as ConML is intended for meta-learning generalization.
>
> We have provided the effect of ConML under distribution shift in Appendix E.
>
> Here, to further evaluate the effect of task interference, we conduct experiments where a portion of tasks during meta-training was 'poisoned' by randomly disturbing the labels of all samples (support and query) independently.
> After meta-training with 10/20%-poisoned task set, we report meta-testing results on 5-way 1-shot MiniImageNet, and calculate relative performance difference (Relative Diff) as (Clean − Poison)/Clean. Results below show that the relative performance decrease is significantly smaller when using ConML. This shows that ConML improves the meta-learning robustness under noisy or corrupted data.
>
> Acc (\%) | Clean Acc | 10%-poison Acc (Relative Diff.) | 20%-poison Acc (Relative Diff.)
> -|-|-|-
> MAML w/o ConML  | 48.7 |45.5(-6.5%)| 37.8(-22.4%)
> MAML w/ ConML  | 56.2 | 54.7(__-2.6__%)| 51.9(__-7.6__%)
>
> ## Q2. ConML's effect when task differences are subtle or overlapping
>
> We would like to discuss by decomposing ConML into optimizing inner- and inter-task distance.
>
> Optimizing inner-task distance benefits generalize to smaller support-size (as shown in Appendix E), would not likely to be effect by task-distribution.
>
> Optimizing inter-task distance helps generalize to ood tasks (as shown in Appendix E). If the meta-training task differences are subtle or overlapping, generalizing to ood tasks becomes more important. This is because if not testing on ood tasks, we can use supervised learning (maybe a little extreme, but the effect of meta-learning itself degrades).
>
> ## Q3.Computational overhead for optimization-based meta-learners.
> In our experiments,
> we only use the weights in the last layer (linear classifier) as representation of optimization-based meta-learners.
>
> This is a very rational concern. We choose model representation by the ‘essential/influential minimal set’ (EMS) of parameters that explicitly individualises the models. And for optimization-based meta-learners, they are inner-updated model weights.
> For meta-learners with very large EMS, we can selectively choose a subset.
> For instance, in MAML models that inner-updates all parameters, limited computational budgets may restrict us to contrasting only the parameters of the last few layers—this approach still achieves comparable performance.
> We also found using parameters solely from earlier layers fails, likely because the final layers capture more task-specific information. We would add illustration in our manuscript.
>
> ## Q4.Discrepancy between MAML and other optimization-based meta-learners
> In Section 3.3, we select one representative method for each category of meta-learners as example to illustrate how to get model representation. The principle of choosing the 'essential/influential minimal set' is shared in the category.
>
> In line 198, we used MAML as example to show the principle for optimization-based is contrasting inner-updated model weights, which also applies to FoMAML and Reptile in Table 1. We have also provided detail algorithm procedure for incorporating ConML with each specific meta-learner in Appendix C.
>
>
> ## Q5. Metric-based algorithms' performance
> We assume the fact that Metric-based algorithms slightly down-perform optimization-based ones on miniImagenet, is likely due to their reliance on fixed embeddings functions which may not adapt as flexibly. Note that in Table 1, all metric- and optimization-based algorithms use the same Conv4 backbone as feature extractor, which is a relatively small model that would benefit from task-specific adaptation. Optimization-based ones adapt the backbone but metric-based ones fix it inside each task.
>
> Indeed, metric-based naturally suited for classification tasks.
> This can also be observed by the strong performance of P>M>F in Table 3, which uses a more deep feature extractor and protonet as classifier. Note that it departures from standard protonet by additionally fine-tuning using support set to achieve task-specific adaptation, which might be an evidence of above explanation.
>
> ## Q6. Explicitly clarify the relationship between in-context learning and meta-learning.
> Thanks for your suggestion. In fact, we have discussed the relation between ICL and meta-learning and reformulate training ICL model as meta-training in Appendix D.
> Apart from these, we have also noticed
> a recent paper [1] exactly answers this with "ICL model is meta-learner with minimal inductive bias", we would add this to our manuscript.
>
> [1]Why In-Context Learning Models are Good Few-Shot Learners? ICLR2025
>
> ## Q7. Related works
> We find the two works provided by the reviewer are highly relevant and very good works. We certainly would add them to references.

---

> > ### Comment · Reviewer_euqW · 2025-08-04
> >
> > Thank you for your responses.
> >
> > - I believe there may have been a misunderstanding regarding Q2 on ConML’s performance when task differences are subtle. This concern is similar to W4 from Reviewer Kgc2, but the response differs significantly.
> > My point remains critical: contrastive learning relies on clear distinctions between negative samples. However, in your setup, tasks are drawn from the same distribution $p^B(\tau)$ (as in Algorithm 2), meaning inter-task differences are small. Using such similar tasks as negative pairs may weaken the contrastive signal, raising concerns about the method’s validity.
> >
> > - Regarding Q4, the line 198 $g(\mathcal D;\theta)=h(;\theta-\nabla_\theta\mathcal L(\mathcal D;h(;\theta)))$ is inconsistent with the Table 1(row 2 column 3) $g(\mathcal D;\theta)=\theta-\nabla_\theta\mathcal L(\mathcal D;h(;\theta))$.

---

> ### Author Response · Authors · 2025-08-04
> **Thanks for your response**
>
> Thanks for your response. We apologize for previous misunderstanding.
>
> **Q2**. This is a very rational concern. We would like to address it in two perspectives.
>
> First, though the scenario raised by the reviewer that "contrasting similar tasks as negative pairs" worths special consideration indeed, it is not a special issue faced by ConML, but a common issue faced by general contrastive learning framework. For example, the widely studied standard (sample-level) contrastive learning [0] also draw samples from a i.i.d distribution $p^n(x)$, which could be similar but treated as negative pairs. But such contrastive learning generally works fine, which could possibly be explained as we are optimizing the **relative** distance where positive pairs should be much similar than negative pairs, and statistically if $\tau_1,\tau_2 \sim$ a continuous $p^2(\tau)$, then $P(\tau_1=\tau_2)=0$.
>
> Second, indeed we can design better sampling strategies and we were taking it as future directions. As we have discussed in the manuscript, we mainly intend to provide the idea and investigate the role of contrastive meta-objective, and there are still many directions to further improve the performance, e.g., contrastive function, distance metric, inner-(subset) and inter-(batch)task sampling strategy.
>
> The mentioned problem involves the inter-(batch)task sampling strategy we have discussed. Optimizing inter-task distance with the current random task batch sampling can effect sub-optimally in scenarios pointed out by the reviewer. It just works statistically. This might be improved by advanced task batch sampling strategy with task-level diversity/difficulty/similarity awareness and schedule, which have been studied in literature, e.g., [1,2,3].
>
> [0] Understanding contrastive representation learning through alignment and uniformity on the hypersphere. ICML 2020
>
> [1] Adaptive task sampling for meta-learning. ECCV2020
>
> [2] The effect of diversity in meta-learning. AAAI2023
>
> [3] Towards task sampler learning for meta-learning. IJCV2024
>
> **Q4**. Thanks for your detailed reading. Table 1 is describing the meta-learner's behavior, where optimization-based learners are updating model parameter by gradient-descent, and thus to introduce how we obtain model representation. Line 198 is the output of optimization-based meta-learner: a predictive model $h$ with updated parameter, to better align our notations $g$ and $h$. We apologize for the confusing and will modify for better clarity in later version.

---

> > ### Comment · Reviewer_euqW · 2025-08-05
> >
> > Thank you for the responses. My concerns are addressed, and I will raise my score.

---

### Official Review · Reviewer_VjCq · 2025-07-03

**Clarity:** 3
**Significance:** 3
**Originality:** 3
**Rating:** 5
**Confidence:** 4

**Summary:**

The authors present ConML, a new way to include a contrastive objective based around model representations as task identity to improve meta-learning methods. Results are obtained across typical few-shot image classification benchmarks (both within and cross-domain) for a number of popular approaches, and a set of synthetic in-context-learning tasks – consistently leading to improvements compared to baseline methods.

**Questions:**

**[Q1]:** “Inner-task sampling K=1” used for few-shot image classification results; If I understand this correctly, the authors only use *one* sample subset in addition to the original one for the contrastive objective;
$\textrightarrow$  This feels very weak and could easily provide almost no additional information, in case a sample that is close to the class-mean would be chosen (especially because only the training data seems to be used).
$\textrightarrow$  Nevertheless, the results indicate it does have quite a significant effect. $\textrightarrow$  I’d like to know the authors’ experiences on this, and potential insights for which cases or applications one might need to choose a higher K.

**[Q2]:** How did the authors choose the representation of each ‘model identity’ in the model space (including ‘mapping function’), e.g. the set of updated parameters for optimisation-based methods? Is the ‘essential/influential minimal set’ that explicitly individualises the models always a good starting point (e.g. param for optim-based, steering-vector for hypernetworks, etc.)? Some details, like concatenation of prototypes for metric-based ones, don’t necessarily seem as obvious;
$\textrightarrow$  Some advice/guidance might be interesting to the reader in case they want to extent this approach beyond the methods used in this work.

**[Q3]:** Less a question but more a weakness: The grammar of the manuscript needs further improvement, and I’d highly recommend the authors check and refine many of the passages. I am aware that this is language-dependent, but given that many tools and/or LLMs are quite well-suited for this task, it shouldn’t be too hard to fix.

Some examples (not exclusive!):
- l.29: “All these works are all under..” -> remove one ‘all’
- l.30: “the the same ” -> remove one ‘the’
- l.42f: “from different task even some of their..” -> from different taskS even IF some..
- l.104: “Details […] is provided” -> ‘are’ provided
- l.137: ‘tau’ after “task” is missing
- l.185: “can exactly serves” -> ‘can exactly serve’
- and several more… (including many missing articles);

---

**Further minor comments and suggestions:**
Since the experiments are all targeting few-shot applications (especially classification), it could be worth extending the literature review/related works to include other few-shot works that use contrastive objectives (some of which use meta-learning in their fine-tuning stage) -- such as:
- Using a contrastive objective to learn class prototypes treated as anchor points to create task-adapted embedding spaces *[Discriminative Sample-Guided and Parameter-Efficient Feature Space Adaptation for Cross-Domain Few-Shot Learning, Perera et al., CVPR2024]*
- Using intra- and inter-task similarity to determine important areas in the exemplar images to aide few-shot classification, optimised via meta-finetuning *[Rethinking Generalisation in Few-Shot Image Classification, Hiller et al., NeurIPS2022]*
- Demonstrate the use of the contrastive method simCLR to enable discrimination between dataset samples while avoiding negative effects due to the supervision signal for few-shot scenarios *[CrossTransformers: Spatially-Aware Few-Shot Transfer, Doersch et al., NeurIPS 2020]*
- And likely others..

---

**TL;DR:** I think the paper provides an interesting view on meta-learning that differs from others, and the experimental results provide evidence regarding its effectiveness as well as general applicability -- hence my rating. (I do, however, expect the authors to rework/improve the language/grammar of the manuscript!)

**Ethical Concerns:**

["NO or VERY MINOR ethics concerns only"]

**Final Justification:**

Having re-read parts of the paper, the authors' rebuttal and the other reviewers' questions, I still believe this paper is a valuable addition to the community -- especially since meta-learning hasn't received as much attention lately, and the authors provide an interesting new way of thinking about meta-learning in the context of various different few-shot methods.

I do agree with the concerns of reviewer *Kgc2* in terms of memory increase (which is a concern), and potentially the robustness (although one or very few datasets not showing major improvements is, at least to me, more related to honesty w/o excessive finetuning than to lack in robustness, given that the authors demonstrate their findings on many different datasets).

$\rightarrow$ However, I do think this works contributes a valuable addition as stated above, and will therefore stick with my initial rating and recommend acceptance.

*P.S.:* Checklist requires major rework, as detailed in the *Paper Formatting Concerns* section.

**Limitations:**

Some limitations are briefly included within (e.g. 4.3.2), but no 'proper' discussion. I'd recommend the authors think about potential limitations their approach could face to provide the reader with some insights into where potential difficulties might arise!

**Paper Formatting Concerns:**

**Major concerns with checklist**: The justifications in the Checklist require some major rework – many questions answered with ‘yes’ lack any justification: E.g. 13 justification simply “yes” – and for most others, the authors have simply copied the question and changed the question mark to a dot;
Note: The idea is to justify the answer by pointing to sections in the paper and/or providing additional info, not to provide text that is entirely ‘default’.

Apart from that, no formatting concerns.

**Quality:**

4

**Strengths And Weaknesses:**

## Strengths
**Originality & Significance:**
- The motivation of using a contrastive objective is rather straight-forward and hence easy to understand, and the paper makes a clear effort to guide the reader along step-by-step
- While contrastive objectives have been used frequently in the few-shot domain (including methods that leverage meta-learning approaches, with some mentioned in the references section), the authors propose a different strategy that is novel and seems to consistently improve upon baselines

**Quality:**
- Results provided for popular few-shot benchmarks, including both in-domain and cross-domain classification
- A range of additional results and insight provided in the appendix, e.g. ConMLs impact on different architectures

**Clarity:**
- The paper is mostly easy to follow, and the use of algorithmic pseudocode further supports a clear interpretation of the underlying concepts

## Weaknesses
- The choice of how a ‘model identity’ is represented is somewhat obscure and simply provided as ‘defined’ without much explanation or reasoning/insight – could be improved to provide a basis for readers that might want to transfer the concept to different methods
- A significant number of typos and grammatical errors that should be corrected

---

> ### Author Rebuttal · Authors · 2025-07-31
>
> We are very glad, and appreciate it a lot to meet such a detailed and professional review.
>
> ## Q1. Inner-task sampling times K=1
> First we want to emphasize the meaning of K=1. Though the reviewer might have understood it correctly, the expression in question "only use one sample subset" might be a little confusing. It does not mean that the subset contains K-shot per class. It means in each episode, we sample K subsets from, and to contrast with, the complete set ($D_\tau^{tr}\cup D_\tau^{val}$), following certain sampling strategy $\pi_\kappa$. With the specification in line 254, in each episode we are contrasting model representation obtained from $D_\tau^{tr}$ with $D_\tau^{tr}\cup D_\tau^{val}$.
>
> We acknowledge the concern about the potential weakness of using only one subset (K=1) for the contrastive objective. In our experiments, Our choice of K=1 was motivated by computational efficiency. However, we agree that increasing K could provide more robust supervision. We observed no significant performance gains when increasing K from Table 4. We infer this is because the large meta-training episode (iteration/step) number, generally over $10^4$, with randomly sampled tasks/subsets, mitigates and compensates for the effect of K in each episode. We would need a larger K for meta-training with only fewer episodes.
>
> ## Q2. Model identity representation
> The ‘essential/influential minimal set’(EMS) that explicitly individualises the models is a very nice summarization!
> For meta-learners with very a large EMS, we can selectively choose a subset.
> For instance, in large MAML models that update all parameters, limited computational budgets may restrict us to contrasting only the parameters of the last few layers—this approach still achieves comparable performance.
> However, using parameters solely from earlier layers fails, likely because the final layers capture more task-specific information.
>
> We believe that the alignment between the EMS's role in the meta-learner and the chosen distance metric is crucial for success.
> For example, we can find that euc distance performs much better than cosine in ProtoNet (Fig.3),
> since ProtoNet classifies using Euclidean distance and ConML contrasts the classifier's weights—capturing the learned representations more accurately.
> Although cosine works generally, it would be interesting to tailor distance metrics for various parameter types (e.g., classifiers, MLPs, CNNs, GNNs, Transformers).
>
> ## Q3. Grammar and typos
> We apologize for the grammatical errors and typos. We will thoroughly proofread the manuscript and use LLM-based tools to ensure clarity and correctness in the revised version.
>
> ## Q4. Related works
> We will expand the related works section to include the suggested contrastive learning papers (Perera et al., Hiller et al., Doersch et al.).
>
> ## Q5. Proper discussion of limitations
> We will modify to add a formal "Limitation" section for proper discussion, including current limitation mentioned in sec 4.3.2 sec 6.
>
> ## Q6. Checklist
> Thanks for your kind reminding. We will modify and expand the justifications.

---

> ### Comment · Reviewer_VjCq · 2025-08-04
> **Thank you for the response**
>
> I'd like to thank the authors for the responses to my questions -- all of them have been appropriately addressed.
>
> I will decide on my final rating once the other reviewers' rebuttal answers have been addressed and discussed.

---

### Decision · Program_Chairs · 2025-09-17

**Decision:**

Accept (oral)

**Comment:**

This paper proposes ConML, which learns models for a variety of tasks by performing meta-learning based on the idea of contrastive learning. In particular, by using task identity as a new source of supervision and carrying out contrastive learning not in representation space but in model-parameter space, the authors succeed in theoretically deriving an excess risk bound that is independent of the concrete problem setting and the learner of the meta learning. As a result, the theory applies in a unified way not only to common meta-learning paradigms such as optimization-based, metric-based, and amortization-based methods, but also to the so-called in-context learning (ICL). The numerical experiments showed the consistent improvement by the proposed contrastive approach.

While most reviewer concerns were largely resolved through discussion, the robustness of the empirical results and the additional computational cost introduced by the proposed method remained primary points of focus. Regarding robustness, the authors provided additional experiments across various datasets. As for computational cost, a concern remains that memory overhead grows rapidly with the number $K$ of subsampled datasets used to evaluate distances between models within the same task. However, the authors report from experiments that $K=1$ yields sufficiently good results, which serves as a reasonable counterargument.

The idea of performing contrastive learning in model-parameter space is highly interesting and offers a new perspective that bridges contrastive learning and meta-learning. Consequently, the paper provided the universally applicable risk bound and presented results that extend to the increasingly prominent ICL. Considering this as a meaningful contribution to the NeurIPS community, I recommend the paper for an oral presentation.